# Precipitation Variability for Protected Areas of Primary Forest and Pastureland in Southwestern Amazônia

Rodrigo Martins Moreira [1], Bruno César dos Santos [2], Rafael Grecco Sanches [2], Vandoir Bourscheidt [3], Fernando de Sales [4], Stefan Sieber [5,6,*] and Paulo Henrique de Souza [7]

1 Geomatics and Statistics Laboratory, Department of Environmental Engineering, Federal University of Rondônia, Porto Velho 76801-974, RO, Brazil
2 Department of Hydraulics and Sanitation, University of São Paulo, São Paulo 05508-010, SP, Brazil
3 Department of Environmental Sciences, Federal University of São Carlos, São Carlos 13565-905, SP, Brazil
4 Department of Geography, San Diego State University, San Diego, CA 92182, USA
5 Leibniz-Zentrum für Agrarlandschaftsforschung (ZALF) e. V., 15374 Müncheberg, Germany
6 Department of Agricultural Economics, Humboldt Universität zu Berlin, 10099 Berlin, Germany
7 Department of Geography, Federal University of Alfenas, Alfenas 37130-000, MG, Brazil
* Correspondence: stefan.sieber@zalf.de

**Abstract:** Daily and monthly rainfall data provided by surface rain gauges in the Amazon Basin are sparse and defective, making it difficult to monitor rainfall patterns for certain portions of its territory, in this sense, estimations of precipitation from remote sensing calibrated with rain gauge data are key to overcome this problem. This paper presents a spatiotemporal analysis of the precipitation distribution for Rondônia State, in southwestern Amazonia. Data from Climate Hazards Group InfraRed Precipitation and Station (CHIRPS) were analyzed, using a pooled time analysis of a forty-year period (1981–2020). Data obtained from remote sensing were validated by rain gauges distributed over the study region. Pixel-by-pixel trend analyzes were developed by applying the Mann-Kendall test and Sen's slope test to study the magnitude of the trend. The analysis revealed that CHIRPS presents a tendency to underestimate precipitation values in most cases. Among the metrics, mean values between very good ($<\pm15\%$) and good ($\pm15$–$\pm35\%$) were observed using PBIAS; mean RMSE values range from 57.8 mm to 107.9 mm; an average agreement level of 0.9 and an average SES of 0.5; and good fit for the linear regression model (average $R^2 > 0.70$) for about 64.7% of the stations. Sen' slope spatialization results show a reduction of approximately $-15$ mm year$^{-1}$, with decrease mainly in the Northern Region of Rondônia, which has extensive areas where the native forest has been replaced by pasture.

**Keywords:** Google Earth Engine; Sen's slope; seasonality; Mann-Kendall

## 1. Introduction

The dynamics of precipitation has outstanding relevance in climate studies in its various spatial and temporal scales. For this reason, the information used in analysis must not be compromised or incomplete, especially with regard to spatial distribution and temporal sequence. Bearing it in mind, several researchers are beginning to look to orbital technology for the necessary help to overcome many kinds of rain gauges flaws. Due to the lack of data in several parts of the Brazilian territory as a consequence of the small number of surface meteorological stations installed, and the temporal gaps they present, analyzing precipitation variation is a challenge [1]. Nearly half of the world's unaltered tropical evergreen forest and sizable tracts of tropical savanna are found in the Amazon Basin. Short-term field observations indicate that these ecosystems are significant global carbon sinks since forests make up a similar percentage of the carbon stored in land ecosystems and contribute roughly 10% of the world's terrestrial primary production. But because El Nio episodes have been occurring more frequently in recent years, tropical terrestrial

ecosystems have seen significant interannual climate fluctuation [2]. For hydrometeorology and climatology investigations, precise long-term estimates of rainfall at fine spatial and temporal resolution are essential, but these data are frequently unavailable in remote areas. Over the Brazilian state of Rondônia [3], found a good precision of satellite-based precipitation products with data from 1981 to 2019 using the Climate Hazards Groups Infrared Precipitattion with rain gauge data (CHIRPS). Also using CHIRPS [4], examined how often extreme rainfall events occurred in the Madeira river basin, the largest Amazon sub-basin. The historical time series analyzed by researchers showed that the basin's average maximum daily rainfall displayed variations that ranged from 30 to 300 mm day1. It should be mentioned that the El Nio/La Nia climate phenomenon has a significant impact on the area under study. Additionally, according to the scale of water balance observation, the results of the trend assessment showed a decrease in the intensity of intense rainfall in study area regions devoid of vegetation cover. Rainfall data sets of sufficient quality and length are frequently absent or just partially present in the State of Rondônia. Thus, to complete databases, rainfall gaps must be filled. [1] analyzed satellite products to fill missing data. To fill in the gaps from 164 rainfall gauge stations in the Amazon basin, satellite rainfall estimates were tested. With $R^2$ ranging from 0.383 to 0.844, large discrepancies between rainfall data were seen; the lowest rainfall locations produced the best results. Additionally, with r and d values greater than 0.899 and 0.950, respectively, the products' best performance was demonstrated during the dry period. The region's most well-represented product, CHIRPS, has the lowest monthly values of mean absolute error (0.979 mm) and root mean square error (3.656 mm). The results show that, despite instances of data overestimation and underestimate, the satellite estimates accurately capture the seasonal variation in rainfall in the region.

Currently, CHIRPS (Climate Hazards Group InfraRed Precipitation with Station data), TRMM (The Tropical Rainfall Measuring Mission), GCMM (Global Precipitation Climatology Centre), CHELSA (Climatologies at High Resolution for the Earth's Land Surface Areas) and CMORPH (Climate Prediction Center—CPC E Morphing Technique—MORPH) represent part of the fundamental technological cast that helps in the monitoring of hydrological cycle variables (such as precipitation), and studies on climate change [5–8]. Advances in remotely sensed products' contribution are key for climate sciences, since several studies confirm that precipitation has shown alterations in its spatial and temporal patterns [3,4,9,10], precipitation in the Amazon Basin is determined by the regularity caused by the action of the Continental Equatorial Air Mass, which originates in the west of the region; the Intertropical Convergence Zone, developed by the convergence of trade winds and cold fronts from extratropical latitudes that frequently impact the southern region of the Amazon; and systems such as the Bolivian High, that act at higher levels of the troposphere [11]. Such oscillations are identified both by the number of episodes and by the rainfall volume; as attested by several climatological studies carried out in South America [12–21].

Daily and monthly rainfall data provided by surface rain gauges in the Amazon are sparse and defective, presenting large periods of missing data, making it difficult to monitor rainfall patterns for certain portions of its territory. The use of remotely sensed data tackles these challenges by providing estimations of precipitations for areas without observations. CHIRPS offers daily data that are estimated using rain gauges. Thus, several studies have used data from CHIRPS to study precipitation in the Amazon, focusing on the validation of daily and monthly precipitation data [9,22,23]. Previous studies that also used the CHIRPS dataset showed that the Amazon Basin's rainfall has increased in some areas, causing more severe droughts and floods. Understanding how precipitation patterns and intensity are changing is crucial since the basin has an impact on the hydrologic and climate cycles on a global scale. Ref. [3] applied several precipitation indices to analyze precipitation variation over the Amazon. Their main findings demonstrate variations in rainfall event timing and intensity at the landscape scale. With an increase of 182 mm in annual rainfall, the western Basin's wet regions have gotten noticeably wetter since 1982. A major drying trend is visible

in the eastern and southern regions, where deforestation is pervasive. Localized changes in precipitation patterns were also noticed. Ref. [24] using CHIRPS dataset examined links between the age of deforestation and spatiotemporal patterns in rainfall between 1981 and 2020. The results demonstrate cohesive correlations between negative dry-season rainfall trends and old-growth deforested areas, despite the presence of significant regional heterogeneity. Up to ten years old deforestation increased rainfall, but longer deforested places had less rainfall during the dry season. These findings imply significant alterations in the Amazon's hydroclimate as well as greater susceptibility to future land cover changes.

With the advancement of computational technological resources and orbital remote sensing, an exciting field of study has emerged, allowing the application of trend analysis on large scales and areas without monitoring by pluviometric stations. And that is precisely one of the main limiting factors for climatological studies in the state of Rondônia [1]. Several studies focused on identifying and analyzing temporal trends use the Mann-Kendall (MK) test, which is a robust and widely applied method [25–27]. Together with MK method, Sen's Slope method (Sen's Slope—SS) is applied, which allows the observation of the trends and magnitude of changes [28,29].

In the midst of these initiatives, a gap is identified regarding studies developed in the southwest region of the Legal Amazon, more specifically in the State of Rondônia, which has increasing deforestation rates. The main gap that this article aims to fill is to provide a time series analysis for protected areas and pasturelands. The federation unit has presented unsustainable patterns of land use and occupation since the 1970s. The main factors for deforestation are the opening and expansion of highways and roads, and intense conversion of native primary forests into pasture. This situation is a potential threat to the balance of the hydrological cycle [30].

Given the relevance of the theme and lack of studies in the mentioned area, this paper is of importance because it presents an analysis of the precipitation spatiotemporal distribution for the state of Rondônia. The study was carried out using 40 years of CHIRPS estimates, which were validated with data from surface rain gauges in a monthly timescale. In order to spatialize the temporal variability of precipitation, an analysis of the precipitation trend was developed pixel-by-pixel, in an annual scale based on the nonparametric statistical tests of Mann-Kendall and Sen's slope.

The objective of the researchers was to present a spatial-temporal trend analysis of precipitation for primary forest protected areas and non-protected areas.

## 2. Materials and Methods

### 2.1. Study Area Description

The area of the study consists of the state of Rondônia, located in the Southeast Region of the Amazon biome (Figure 1). It is in the North Region of Brazil, and it is part of the Legal Amazon. We selected the geopolitical boundary due to the State's lack of studies regarding trends in precipitation over protected and non-protected areas. The State of Rondônia is a great case study for deforestation, since the 1980's, when we have the start of Landsat 4 mission, it is notable the conversion of primary forests to pasture.

According to the Brazilian Institute of Statistics and Geography (Instituto Brasileiro de Geografia e Estatística, in Portuguese—IBGE), the state of Rondônia has 237,765.347 km², 1.7 million inhabitants, and 52 municipalities [31]. The main cities are: the capital Porto Velho, Ji-Paraná, Ariquemes, Cacoal and Vilhena [32]. As reported by Köppen-Geiger climate classification system (1928), the territory's climate is divided into two types: tropical rainforest with a well-defined dry season (Aw) and tropical rainforest with a short dry season (Am). Both climate classifications have an average rainfall between 1300 and 2600 mm/year, which is more intense in December, January and February. The dry season occurs in June, July and August. The average temperature varies between 24 and 26 °C [11].

The state of Rondônia presents terrains with homogeneous altitudes without major irregularities. Approximately 94% of its territory is between 100 and 600 m above sea level. Figure 1 shows the Köppen-Geiger climate classification system provided by Alvares

et al. [33] for the state of Rondônia. The classification of land use, made available by MapBiomas [34], is highlighted.

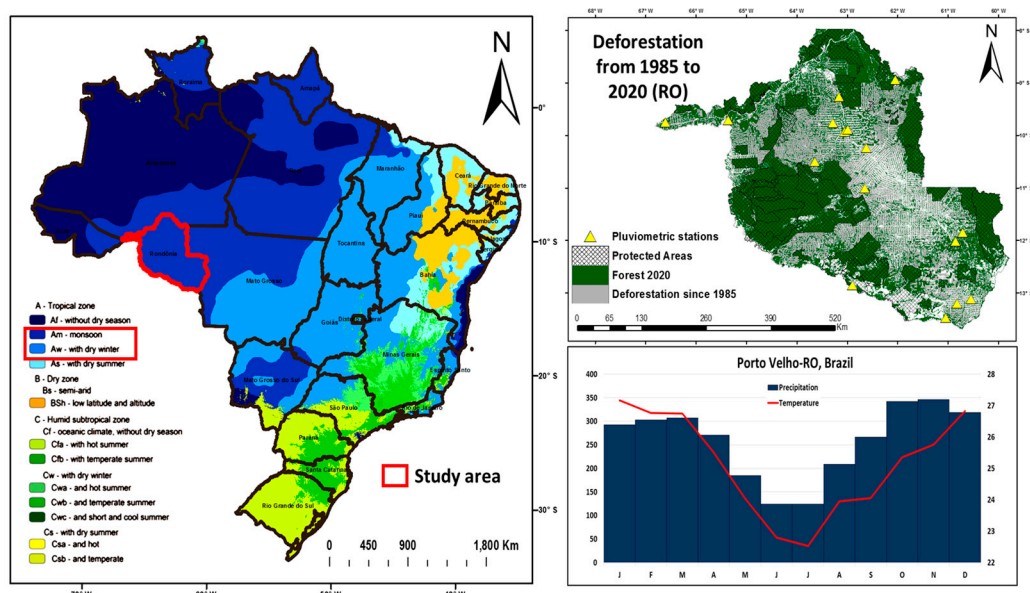

**Figure 1.** Study area presenting Köppen-Geiger climate classification; the location of rain gauges and classification of land use and occupation for the year 2020; and the loss of primary vegetation from 1985 to 2020; and the climatological normal (temperature and precipitation) from the capital of Porto Velho-RO.

During the summer, atmospheric circulation exhibits an important pattern through the trade winds that transport moisture from the Tropical Atlantic to the Amazon region. Added to this, at this time of the year, due to the advances of cold fronts coming from the southern portion of Brazil, a thermal low over the Chaco region, are associated with intense convective activities, increasing cloud cover over the Central Amazon region and, therefore, providing conditions for the SACZ to be more active and intense on rainfall in the southern and western regions of the Amazon [35].

Rainfall seasonality is an important feature, as it allows understanding of interannual variability and extreme rainfall events. Changes in these rainfall patterns can cause impacts due to natural causes or anthropic processes [16,35]. In this sense, interannual variability depends on the fields of sea surface temperature (SST) anomalies in the Pacific or Tropical Atlantic at the beginning and end of the rainy season. The combination of anomalous atmospheric circulations, influenced by SST, affect the positioning of the ITCZ over the Atlantic and, therefore, influencing the rainfall distribution over northern South America. Therefore, El Niño and La Niña events cause an increase or decrease in rainfall anomalies in the Amazon [35,36].

According to [37], the increase in deforestation in recent decades in the southern region of the Amazon has generated some changes in land use and occupation, which may influence the transition period from the dry to the wet season in Rondônia, due to the decrease in forest cover in the region. region, where changes in surface radiation balance caused by deforestation are the likely drivers of changes in rainfall patterns in the rainy season. In this sense, although changes in annual precipitation totals are not significant, the implications of a longer dry season are important for a variety of benefits that forest cover exerts, such as water availability and ecological aspects [37].

### 2.2. Selection of Rain Gauges

The second dataset used in the study was the Climate Hazards Group Infra-Red Precipitation with Station data (CHIRPS) produced by Funk et al. [38]. Data between the years 1981 to 2020 were collected using the Google Earth Engine platform. Two temporal

scales were used: the first one was the monthly scale, employed for data validation; the second one was the annual scale, employed to calculate the pixel-by-pixel trends. The data were processed using the RStudio platform [39].

### 2.3. Collection of CHIRPS Data

Rainfall gauge data was obtained using the National Water Agency (ANA) plugin for the QGis program [40]. The ANA Data Acquisition tool automatically downloads from several pluviometric and fluviometric stations. Using the boundary of the state of Rondônia as an area of interest, 162 stations were first selected. After a first filtering, considering a temporal criterion, only 28 rain gauges had data from 1981 to 2020. This interval was defined due to the CHIRPS data being available since 1981. A new filtering was performed on these 28 stations, selecting only those presenting less than 15% of missing data. After these filtering steps, the researchers selected 16 stations, shown in Table 1.

**Table 1.** Information from rain gauges in Rondônia regarding their identification code, latitude, longitude, percentage of faults and elevation in meters.

| Station Code | Lat | Long | Missing Data (%) | Elevation (m) | Min | Max | CV |
|---|---|---|---|---|---|---|---|
| 862000 | −8.93 | −62.06 | 10 | 80 | 0 | 543.3 | 0.74 |
| 963000 | −9.93 | −63.06 | 8.2 | 114 | 0 | 591.1 | 0.86 |
| 963001 | −9.26 | −63.16 | 2.6 | 98 | 0 | 551.3 | 0.84 |
| 963004 | −9.89 | −62.99 | 5.8 | 113 | 0 | 687 | 0.85 |
| 965001 | −9.70 | −65.36 | 10.8 | 98 | 0 | 652.8 | 0.98 |
| 966000 | −9.76 | −66.61 | 4.5 | 148 | 0 | 532.9 | 0.94 |
| 1062002 | −10.24 | −62.63 | 3.1 | 220 | 0 | 566.1 | 0.88 |
| 1062003 | −11.00 | −62.66 | 10.7 | 251 | 0 | 467.7 | 0.80 |
| 1063000 | −10.51 | −63.65 | 10.1 | 187 | 0 | 724,6 | 0.87 |
| 1063001 | −9.76 | −63.29 | 11.3 | 115 | 0 | 588.2 | 0.82 |
| 1160000 | −12.02 | −60.86 | 5.1 | 264 | 0 | 505.9 | 0.96 |
| 1160001 | −11.85 | −60.72 | 9.6 | 269 | 0 | 634.3 | 0.85 |
| 1262000 | −12.85 | −62.90 | 6.3 | 158 | 0 | 663.7 | 1.04 |
| 1360000 | −13.11 | −60.55 | 11.2 | 419 | 0 | 524.1 | 0.84 |
| 1360001 | −13.20 | −60.82 | 9.2 | 257 | 0 | 658.7 | 0.97 |
| 1360002 | −13.48 | −61.05 | 7.6 | 171 | 0 | 848.4 | 1.17 |

### 2.4. Metrics for Validating CHIRPS Data

The validation of CHIRPS data was performed by superimposing the rain gauge location point over the CHIRPS image pixel. The time scale used for the tests was monthly. In order to validate the CHIRPS data, the coefficient of determination $R^2$ (Equation (1)) was used, whose result varies between 0 and 1. The closer to 1, the more accurate is the representation of the values obtained by CHIRPS, being a value of 1 a perfect correlation between the data sets (Equation (1).

$$R^2 = \frac{\sum_{t=1}^{n}\left[\left(y_t - \overline{y}\right)\left(\hat{y}_t - \overline{\hat{y}}\right)\right]^2}{\sum_{t=1}^{n}\left[y_t - \overline{y}\right]^2 \sum_{t=1}^{n}\left[\hat{y}_t - \overline{\hat{y}}\right]^2} \tag{1}$$

where, $n$ is equivalent to the number of samples; $t$ is the period; $y_t$ is the value observed at the rain gauge in period $t$; $\hat{y}_t$ is the value of the data obtained from analyzes on each cell in period $t$; $\overline{\hat{y}}$ is the mean of the predicted values.

Root-Mean-Square-Error (RMSE) was used as a measurement error between rain gauge data and CHIRPS estimates ((Equation (2)), in which the closer to 0, the most accurate is the adequacy of the CHIRPS values:

$$RMSE = \sqrt{\frac{1}{n} \sum_{i=1}^{n} [Z.(y_i) - z(y)]^2} \tag{2}$$

The percentage bias represents the tendency of CHIRPS values to underestimate or overestimate the values collected by the stations (Equation (3)).

$$Pbias = \frac{\sum_{i=1}^{n} (y_i - S_i)^2}{\sum_{i=1}^{n} y_i} . 100 \tag{3}$$

where $n$ is the number of measured data; $y_i$ and $S_i$ are respectively observed and estimated data at time $i$.

And the index of agreement ($d$), created by Willmott [41], is further used to identify the degree of agreement between the values of the stations and the values obtained by CHIRPS, in which 1 is a perfect agreement (Equation (4)).

$$d = \frac{\sum (y_i - s_i)^2}{\sum_{i=1}^{n} (|s_i - \overline{y}| + |y_i - \overline{y}|)^2} \tag{4}$$

where $y$ is equal to the mean value of the data observed by the stations and $s$ represents the CHIRPS estimates.

Finally, the Nash-Sutcliffe efficiency coefficient was also determined. Varying from infinity to one, it states that the closer to one, the more accurate is the prediction of the simulated data. The coefficient was calculated using the following formula (Equation (5)):

$$NSE = \frac{\sum (y_i - s_i)^2}{\sum (y_i - \overline{s}_i)^2} . 100 \tag{5}$$

where $y_i$ is the observed value and $\overline{s}_i$ represents the mean of the observed value.

The Mean Absolute Error (MAE) provides the mean of the absolute difference between the goal value and model prediction. In MAE, distinct errors are not given a varied weight, but rather, the scores rise linearly as the number of errors rises. The average of the absolute error numbers is used to calculate the MAE score. A mathematical operation called the Absolute turns a negative number positive. As a result, while computing the MAE, the difference between an expected value and a forecasted value can be either positive or negative.

Following is the equation to compute the MAE value (Equation (6)):

$$MAE = \frac{1}{n} \sum_{i=1}^{n} |x_i - x|^2 \tag{6}$$

In recent years, calibration and evaluation of hydrological models have largely relied on the Kling-Gupta efficiency (KGE), which more evenly combines the three components of the Nash-Sutcliffe efficiency (NSE) of estimated errors. When maximizing its value for precipitation estimates, the KGE does not, however, take into consideration reference forecasts or simulations and continues to underestimate the variability of precipitation time series. The KGE can be calculated according to the following equation (Equation (7)):

$$KGE = 1 - \sqrt{(r-1)^2 + \left(\frac{\sigma \, sim}{\sigma \, obs} - 1\right)^2 + \left(\frac{\mu \, sim}{\mu \, obs} - 1\right)^2} \tag{7}$$

where $\sigma$ *obs* is the standard deviation in observations, $\sigma$ *sim* the standard deviation in simulations, $\mu$ *sim* the simulation mean, and $\mu$ *obs* the observation mean.

### 2.5. Historical Analysis of CHIRPS Precipitation Data

For the historical analysis of precipitation data from the state of Rondônia, classification and rainfall trends tools were used. The methods were applied pixel-by-pixel, for a total of 805,737 representative pixels for the State.

The CHIRPS data were obtained in matrix format with Tagged Image File (TIF) extension, including annual precipitation from 1981 to 2020, in a total of 40 images, which were adjusted to the SIRGAS 2000 Geodetic Reference System. In order to process the spatial data, the free programs RStudio [39] and QGis [42] were used.

The calculation for Mann-Kendal test and for Sen's Slope were performed in RStudio. Calculations were made for each pixel of the matrix files.

### 2.6. Standard Precipitation Index (SPI–6)

The Standard Precipitation Index (SPI) was introduced by McKee et al. [43] for the analysis of dry and wet periods. It determines the intensity of each period in different temporal scales, ranging from 1, 3, 6, 9, 12, 24 and up to 48 months. SPI is widely used because it only requires precipitation data as an input variable. Details about the mathematical formulations and statistical procedures for the elaboration of the SPI are found in McKee et al. [43]. For this analysis specifically, the monthly scale was used.

The SPI is based on the probability density function of gamma distribution, where $\alpha$ is the shape parameter ($\alpha > 0$), $\beta$ scale parameter ($\beta > 0$) and $x$ is the amount of precipitation (Equation (8)).

$$g(x) = \frac{X^{\alpha-1} \times e^{\frac{-x}{\beta}}}{\beta^\alpha \Gamma(\alpha)} \; to \; X > 0 \tag{8}$$

where $\alpha > 0$ shape parameter; $\beta > 0$ scale parameter; $x > 0$ the amount of precipitation (mm) and $\Gamma(\alpha)$ the full gamma function.

To estimate the $\alpha$ and $\beta$ parameters of the gamma distribution, we used (Equation (9)):

$$\alpha = \frac{1}{4A}\left(1 + \sqrt{1 + \frac{4A}{3}}\right) e\hat{\beta} = \frac{x}{\alpha} \tag{9}$$

where $A$ is given by (Equation (10)):

$$A = ln\,(\overline{x}) - \frac{\sum_{i=1}^{n} ln(x)}{n} \tag{10}$$

where $\overline{x}$ is the average precipitation, $x$ represents the annual precipitation and $n$ represents the size of the series.

Thus, the cumulative distribution is transformed into a normal probability distribution with a mean equal to zero and a standard deviation equal to one. The cumulative probability of occurrence of each value on the monthly scale is then estimated. For standardization, the value of $Z$ is then calculated, where the precipitation $i$ is subtracted from the average precipitation and divided by the standard deviation (Equation (11)).

$$SPI = Z_i = \frac{(P_i - \underline{P}_i)}{\sigma_i} \tag{11}$$

Then, the values are classified according to Table 2 into dry and rainy periods. Due to the standardization factor, the SPI allows comparing different regions.

**Table 2.** Classification of SPI values.

| SPI Values | Categories |
|---|---|
| $\geq 2$ | Extremely wet |
| 1.50 to 1.99 | Severely wet |
| 1.00 to 1.49 | Moderately wet |
| 0.99 to –0.99 | Near normal |
| –1.00 to –1.49 | Moderately dry |
| –1.50 to –1.99 | Severely dry |
| $\leq -2$ | Extremely dry |

*2.7. Mann-Kendall Test*

For this test, the monthly scale was used for the table creation and the annual scale for the pixel-by-pixel spatialization of the coefficients. The nonparametric trend test used in this study was proposed by Mann [44] and Kendall [45]; therefore, it was applied in order to identify possible trends of precipitation decrease or increase for the State of Rondônia in a period of 40 years. The test is calculated according to Equation (12).

$$S = \sum_{i=2}^{n} \sum_{j=1}^{i=1} signal\left(x_j - x_i\right) \tag{12}$$

where $S$ is described as the result of the sum of $(x_j - x_i)$, in which $x_j$ is taken as the first value after $x_i$, and $n$ is the amount of data for the entire period. Thus, in the peer-to-peer analysis, each pairing will be assigned the values The probability distribution of the $S$ statistic tends towards normality when the number of observations ($n$) of the samples is large, with zero mean and variance calculated from Equation (13):

$$signal = \begin{cases} +1\ if\ \left(x_j - x_i\right) > 0 \\ 0\ if\ \left(x_j - x_i\right) = 0 \\ -1\ if\ \left(x_j - x_i\right) < 0 \end{cases} \tag{13}$$

where $t_p$ is taken as the amount of data with equal values in a given group, $q$ is the number of groups with equal values in the time series in a group $p$.

The Mann-Kendall test is based on the value of the variable $Z_{MK}$. Because $Z_{MK}$ is a two-tailed test, it is assumed as a $p$-value ($-1.96 < Z_{MK} < 1.96$) calculated according to Equation (14).

$$Z_{MK} = \begin{cases} \frac{S-1}{\sqrt{VAR(S)}}\ if\ S > 0 \\ 0\ if\ S = 0 \\ \frac{S-1}{\sqrt{VAR(S)}}\ if\ S < 0 \end{cases} \tag{14}$$

*2.8. Sen's Slope Analysis*

Once significant trends have been identified, it is important to estimate the magnitude of this trend. For the test, the annual temporal scale was used. In the various methods applied for this purpose, the normality of the dataset is a prerequisite, being highly sensitive to outliers. In order to overcome this limiting factor, it is necessary to apply a more robust test adapted to non-parametric data, such as Sen's Slope (*SS*) [46], aimed at identifying magnitudes in time series trends (Equation (15)).

$$SS = Median\left\{ \left[ \left(\frac{x_i - x_j}{i - j}\right)_{j=1}^{j=n-1} \right]_{i=j+1}^{i=n} \right\} \tag{15}$$

where $x_i$ and $x_j$ are pairs at given times $i$ and $j$ ($j > i$), respectively.

*2.9. Laplace Trend Test*

The Laplace test is a measurement that assesses the relationship between the centroid of observed arrival times and the midpoint of the observational period. The standardized normal random variable is roughly represented by this measurement. According to Sanches [18], the Laplace test allows for the design of this kind of evaluation as a function of precipitation, reiterating the tool's potential for these studies in historical data series. The following equation determines the trend based on a given value *u(t)* and taking into account a given period [0, *t*] [47] (Equation (16)).

$$u(t) = \frac{\frac{\sum_{i=1}^{t}((i-1)n_i)}{N(t)} - \frac{(t-1)}{2}}{\sqrt{\frac{t^2-1}{12(N(t))}}} \tag{16}$$

where, *t*: represents the number of months; $n_i$: is the variable analyzed at time *i*; and $N_{(t)}$: indicates the cumulative number in relation to the analyzed variable.

A score above zero indicates an upward or growing trend, whereas a score below zero indicates a downward or decelerating trend. We are at least 95% certain that a substantial tendency toward the positive is present when the score is more than (less than) +1.96 (−1.96). (downward). A score of 0 or very close to it indicates a horizontal trend (has constant rate).

**3. Results**

*3.1. Validation of CHIRPS Data*

Figure 2 presents the total values of the data observed by the stations in relation to those estimated by CHIRPS. In the figure, it is possible to observe the behavior of the two precipitation variables over time at each station location. It was also possible to observe similar behaviors in some moments of the timeseries, but most of them showed differences between the observed values and the estimated ones. Despite this, some rain gauges stations had similarities in the fluctuation of values regarding CHIRPS.

Table 3 introduces the applied metrics values intending to validate the CHIRPS data.

It is possible to observe that the Root-mean-square-error (RSME) varied for all periods compared, from 57.8 mm.month$^{-1}$ for station 1160001 (269 m) to 107.9 mm/year for station 1360002 (171 m). Therefore, the CHIRPS data present average monthly errors above 50 mm for the study area.

The PBIAS values, which indicate the percentage of underestimation and overestimation regarding the value observed by the stations and the values estimated by CHIRPS, varied between −35.4% for station 966000 (148 m) and 5% for station 1063000 (220 m), which is the station the closest to the true value. The PBIAS metric presented values between very good (<±15%) and good (±15% to ±35%), according to Shrestha et al. [48].

As for the NSE coefficient, stations 965001 (98 m) and 1360002 (171 m) presented the lowest value, which was 0.1. The rain gauge station that presented the highest value was 1160001 (269 m) with 0.8, followed by stations 1062001 (143 m) and 1360001 (257 m) with 0.7. Willmott's index of agreement (d) ranged between 0.8 and 0.9.

Finally, the results of the Coefficient of Determination $R^2$ showed, for the vast majority of stations, values ranging between 0.5 and 0.9 and, therefore, indicating an excellent fit. More specifically, for about 64.7% of the stations, the values remained (>0.7), and for the remaining 35.3% these can still be considered satisfactory (>0.5).

In order to spatially visualize the rain gauges that obtained higher or lower performance, Figure 3 presents the spatialization of validation metrics according to each rain gauge.

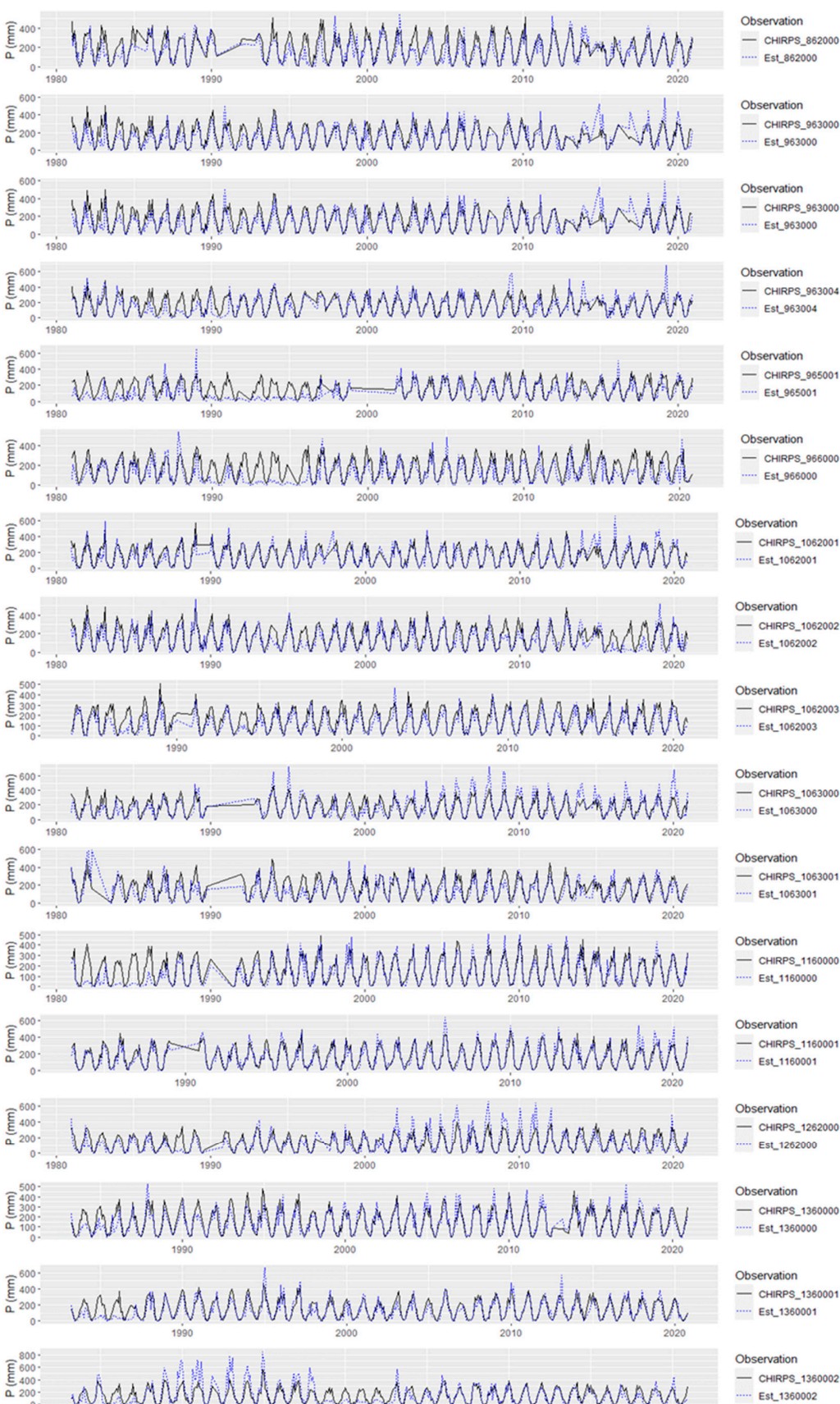

**Figure 2.** Overlay of each rain gauge data and the CHIRPS data for that pixel.

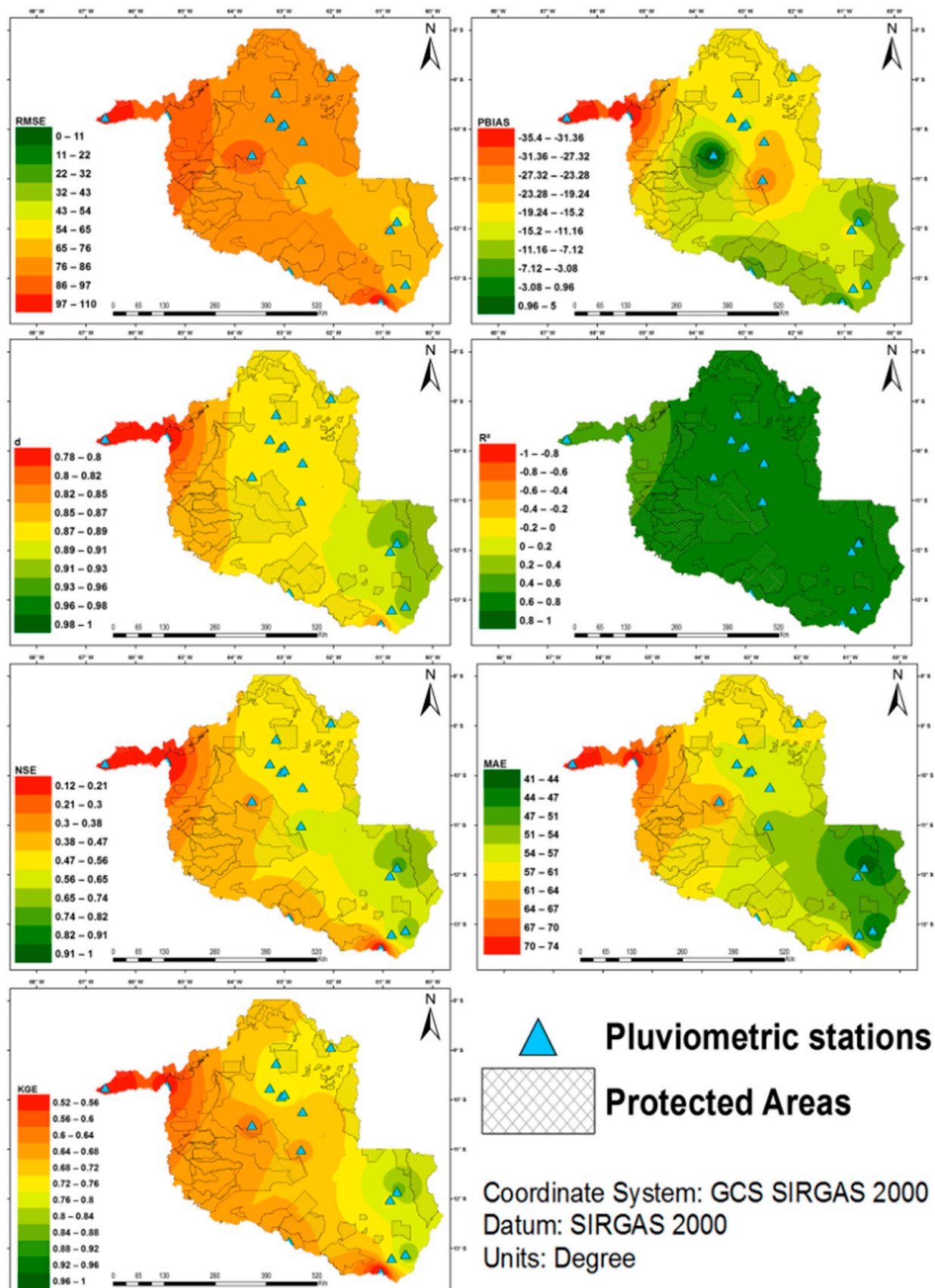

**Figure 3.** Spatialization of metrics RMSE, PBIAS, Willmott coefficient (d), R$^2$, NSE, MAE and KGE.

**Table 3.** Values of the validation metrics of the values estimated by CHIRPS according to the values collected by the rain gauges.

| Station | MAE | RMSE | PBIAS % | NSE | d | $R^2$ | KGE |
|---------|-----|------|---------|-----|---|-------|-----|
| 862000 | 59.03 | 84.33 | −18.2 | 0.56 | 0.88 | 0.66 | 0.73 |
| 963000 | 56.09 | 80.16 | −18.1 | 0.52 | 0.88 | 0.64 | 0.73 |
| 963001 | 58.57 | 84.08 | −17 | 0.52 | 0.88 | 0.63 | 0.73 |
| 963004 | 56.12 | 84.74 | −12.1 | 0.46 | 0.87 | 0.61 | 0.72 |
| 965001 | 71.16 | 95.61 | −33.4 | 0.12 | 0.79 | 0.49 | 0.55 |
| 966000 | 73.8 | 102.65 | −35.4 | 0.16 | 0.78 | 0.48 | 0.52 |
| 1062002 | 54.64 | 77.3 | −22.4 | 0.56 | 0.89 | 0.69 | 0.72 |
| 1062003 | 52.23 | 69.91 | −26.4 | 0.6 | 0.88 | 0.74 | 0.67 |
| 1063000 | 65.24 | 91.47 | 5 | 0.36 | 0.87 | 0.65 | 0.61 |
| 1063001 | 53.7 | 76.03 | −18.1 | 0.55 | 0.89 | 0.67 | 0.74 |
| 1160000 | 51.35 | 78.23 | −19.3 | 0.59 | 0.9 | 0.69 | 0.74 |
| 1160001 | 40.78 | 57.81 | −2.4 | 0.79 | 0.95 | 0.82 | 0.87 |
| 1262000 | 57.52 | 84.94 | −6.1 | 0.38 | 0.87 | 0.61 | 0.66 |
| 1360000 | 43.58 | 62.85 | −7.9 | 0.71 | 0.93 | 0.75 | 0.84 |
| 1360001 | 47.42 | 70.1 | −17.3 | 0.63 | 0.91 | 0.72 | 0.76 |
| 1360002 | 70.9 | 107.9 | −2.2 | 0.11 | 0.84 | 0.59 | 0.49 |

*3.2. Monthly Rainfall Distribution and Standardized Precipitation Index (SPI)*

This section will describe the precipitation regimen for the State of Rondônia, the analysis will be presented for anthropized and primary forests protected areas in the State.

A summary of results for more dissident years are presented in Table 4.

**Table 4.** Descriptive statistics of precipitation (mm.year$^{-1}$) for years 1983, 1998, 2010 and 2015 presenting minimum, medium and maximum values for protected and pasture areas.

| Year | Protected Areas | | | | Pasture Areas | | | |
|------|-----|-----|-----|-----|-----|-----|-----|-----|
| | min | med | max | CV | min | med | max | CV |
| 1983 | 1272 | 1609 | 2262 | 0.12 | 1264 | 1695 | 2285 | 0.11 |
| 1998 | 1325 | 1668 | 2474 | 0.11 | 1316 | 1714 | 2421 | 0.11 |
| 2010 | 1345 | 1778 | 2265 | 0.10 | 1327 | 1845 | 2282 | 0.08 |
| 2015 | 1418 | 1773 | 2046 | 0.07 | 1407 | 1729 | 2045 | 0.8 |

Figure 4 demonstrates the variability of precipitation volume and its classification from 1981 to 2020. The chart illustrates a fluctuation in the precipitation data, that is, moments in the series when rainfall was above the average rainfall pattern (in blue) and, at other times, below the standard (in red). Among the four decades, 2009 is highlighted as exceptionally rainy (>deviations) and 2015 as exceptionally dry (<deviations). It was also possible to notice that, in recent years, SPI values have been classified as dry years. In other words, precipitation is again deviating from the usual pattern for the study area.

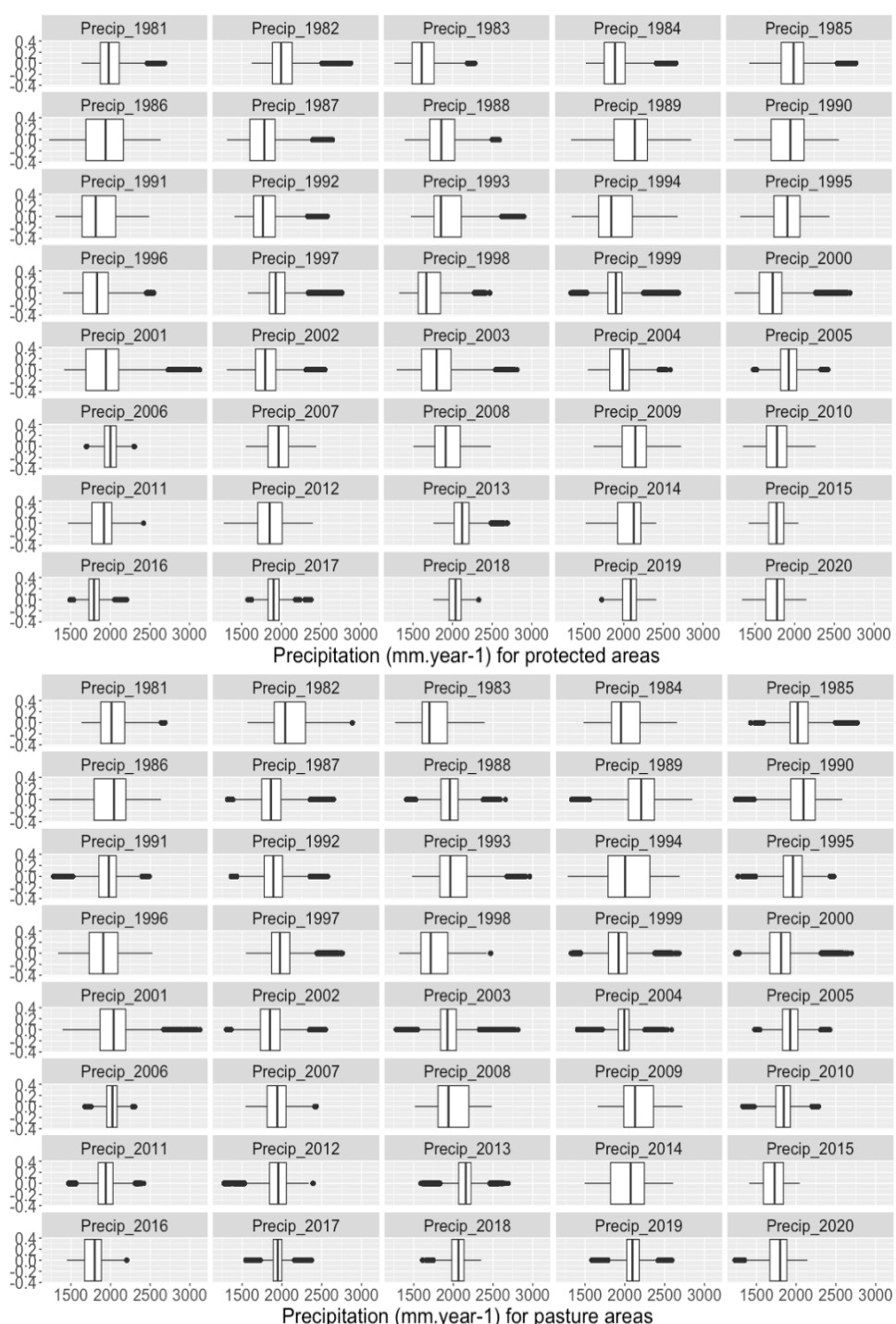

**Figure 4.** Precipitation distribution (mm.year$^{-1}$) for protected areas and pasturelands for years from 1981 to 2020 using CHIRPS data.

Figure 5 demonstrates the spatial and temporal distribution of precipitation over the forty-year-period. It was possible to identify some years in which high precipitation totals were registered, close to or above 3000 mm, and others, with totals below 1500 mm. Figure 5 also illustrates the distribution of the yearly precipitation using the average of the pixels for the entire state in the period from 1981 to 2020, with precipitation data from CHIRPS. The years of 1983, 2010 and 2015 stand out as the driest.

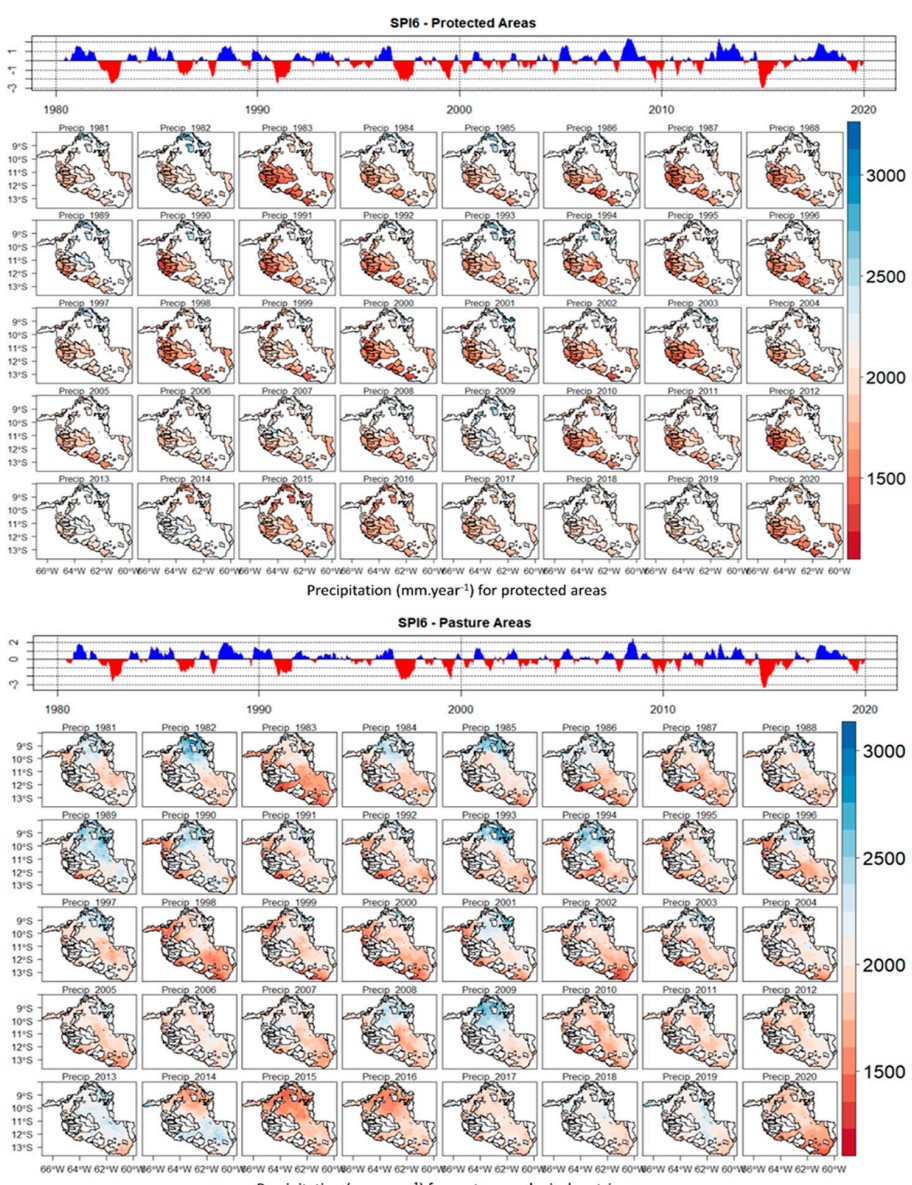

**Figure 5.** Precipitation Spatial distribution (mm.year$^{-1}$) and SPI–6 for protected areas and pasture-lands in the State of Rondônia from 1981 to 2020 using CHIRPS data.

*3.3. Mann-Kendall and Sen's Slope Precipitation Trends Analysis for the State of Rondônia*

Table 5 below lists the results of the trend analysis using the Mann-Kendall non-parametric statistical test for the average monthly rainfall, using the average of the pixels for the entire state of Rondônia, based on the CHIRPS data.

One of the products of the MK test are the $Z_{MK}$ values, which indicate the degree of the tendency to increase, in case the values are positive, or to decrease, in case they are negative. As indicated by the *p*-value, the values were not statistically significant, considering alpha equal to 0.05 and the breakpoints of $Z_{\alpha/2}$ (−1.96 and 1.96). These results are likely influenced by the spatial variations of these indexes, as evidenced in Figure 7 and discussed later. Therefore, despite this result, the values of the Sen's slope are presented, which quantitatively indicates the degree of the trend. It is possible to notice that the months that presented values closer to significance were the months of November ($Z_{MK}$ = −1.43 and SS = −0.52 mm.month$^{-1}$), July ($Z_{MK}$ = −1.1 and SS = −0.07 mm.month$^{-1}$) and January ($Z_{MK}$ = −0.64 and SS = −0.32 mm.month$^{-1}$), with a decreasing trend. On the other hand, values that showed a positive trend are seen in the months of May ($Z_{MK}$ = 0.83 and

SS = 0.27 mm.month$^{-1}$), October ($Z_{MK}$ = 0.71 and SS = 0.3 mm.month$^{-1}$) and August ($Z_{MK}$ = 0.69 and SS = 0.13 mm.month$^{-1}$).

**Table 5.** Result of the Mann-Kendall test for monthly rainfall between 1981 and 2020 with the $Z_{MK}$, Sen's slope (SS) and *p*-value.

| Month | $Z_{MK}$ | SS (mm.month$^{-1}$) | *p*-Value |
|---|---|---|---|
| January | −0.64 | −0.32 | 0.52 |
| February | 0.52 | 0.26 | 0.60 |
| March | 0.52 | 0.34 | 0.60 |
| April | 0 | 1 | 0.001 |
| May | 0.83 | 0.27 | 0.41 |
| June | 0.64 | 0.09 | 0.52 |
| July | −1.1 | −0.07 | 0.27 |
| August | 0.69 | 0.13 | 0.5 |
| September | −0.36 | −0.11 | 0.72 |
| October | 0.71 | 0.3 | 0.48 |
| November | −1.43 | −0.52 | 0.15 |
| December | −0.12 | −0.06 | 0.9 |

Finally, the map in Figure 6 displays the trend of rainfall distribution for the study area. The map shows regions that presented a reduction (red) and others an increase (blue) in the volume of precipitation throughout the year, which possibly affected the global analysis presented above. Among the areas, the southwest portion expressed an increase in the volume of annual precipitation, while the northern portion manifested a reduction in the volume of precipitation.

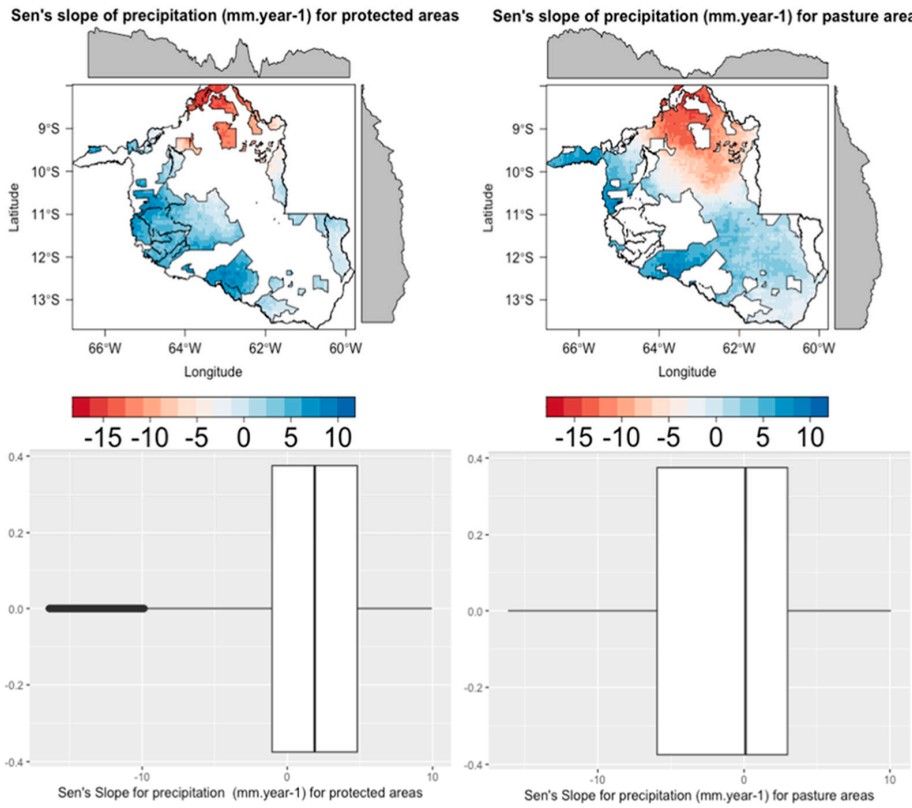

**Figure 6.** Spatialization of the Sen's Slope of magnitude of change in the yearly behavior of precipitation in the state of Rondônia, in which the polygons represent protected areas.

### 3.4. Spatial Analysis of Precipitation Trends in the State of Rondônia Separated per Dry and Rainy Season for Anthropized and Primary Forest Areas

Regarding the twelve months of the year, those with the highest accumulated rainfall were January, February, March and December. The average values for these months were above 200 mm and, therefore, reaffirming that the trimester (January, February, March) is the wettest of the year in the Tropical rainforest climate. Figure 7 registers the Box-Plot of monthly rainfall and seasonal variability with rainfall data from the rain gauge and CHIRPS.

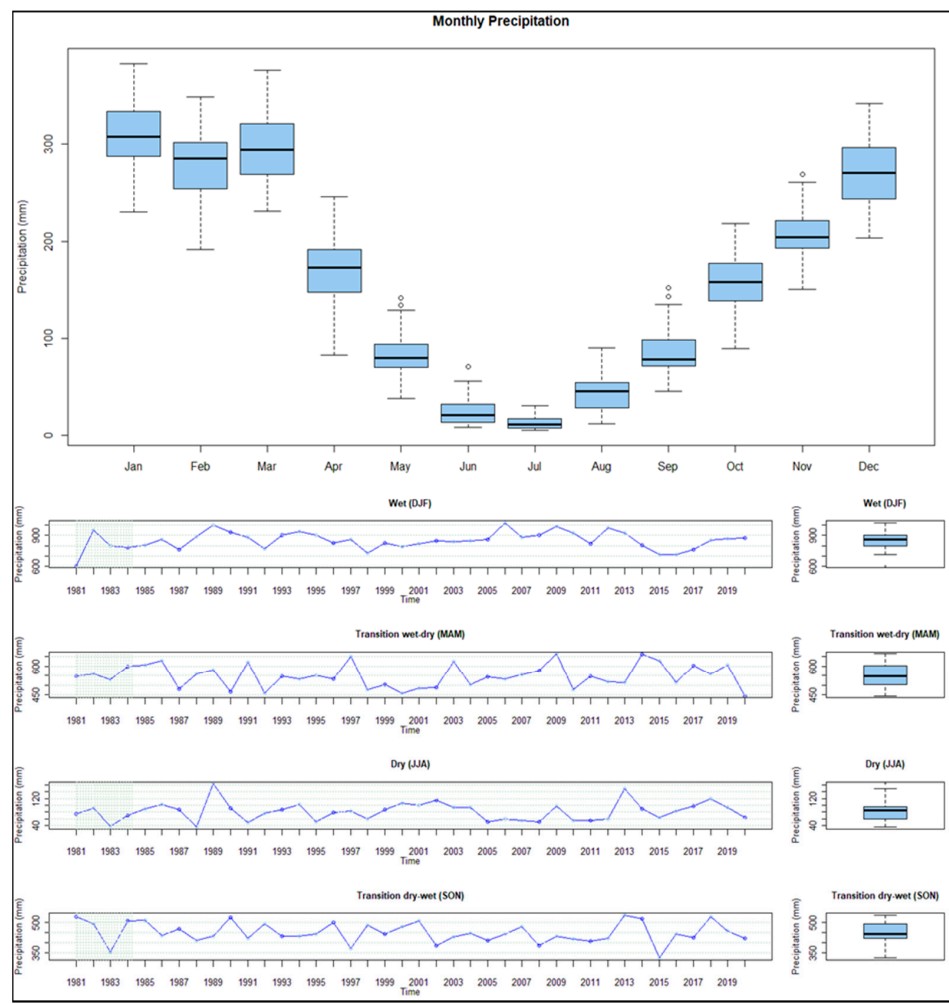

**Figure 7.** Monthly and seasonal distribution of rainfall in Rondônia for the years 1981 to 2020 using CHIRPS data. These are the mean values of all valid pixels of CHIRPS images.

Figure 7 shows a fluctuation over time, with an increasing trend at the beginning of the series. Later on, it goes through some transition periods and, in recent years, decreasing trends. The months of April and October are months of transition from the wet to the dry period and have average precipitation between 100 and 200 mm. Finally, the months of May, June, July, August and September were the months with the lowest average values, below 100 mm. June and July are the driest months. In addition, it is possible to observe that the wettest months (January, February, March, November and December) and transitional months (April and October) have higher precipitation amplitudes in relation to the mean, when compared to drier months (June and July), which have lower amplitudes.

During the rainy season (DJF), there is a small precipitation amplitude between 600 and 900 mm. Throughout the timeseries, it is possible to notice a variability over the years. There are high positive peaks. However, in recent years, mainly for 2012, the volume of the rainy trimester presents a reduction in precipitation (mm).

During the dry-wet transition season (SON) and wet-dry transition season (MAM), the variability showed a greater fluctuation, between 350 to 650 mm over the historical series. And finally, for the dry season (JJA), the variability between 40 and 140 mm showed a few moments with great oscillation in the rainfall accumulated in the driest trimester.

### 3.4.1. Spatial Analysis of Precipitation Trends in the State of Rondônia Separated per Dry and Rainy Season for Anthropized and Primary Forest Areas

In protected primary forests in the rainy season, the years that presented lower values of rainfall with median values approximate to 750 mm.year$^{-1}$ were 1983, 1984, 1992, 1998, 2000, 2002, 2010, 2016 and 2020. For pasture areas in the rainy season, the years that presented lower values of rainfall with median values approximate to 750 mm.year$^{-1}$ were 1984, 1985, 2002, 2007, 2015, 2016 and 2020, as seen in Figure 8.

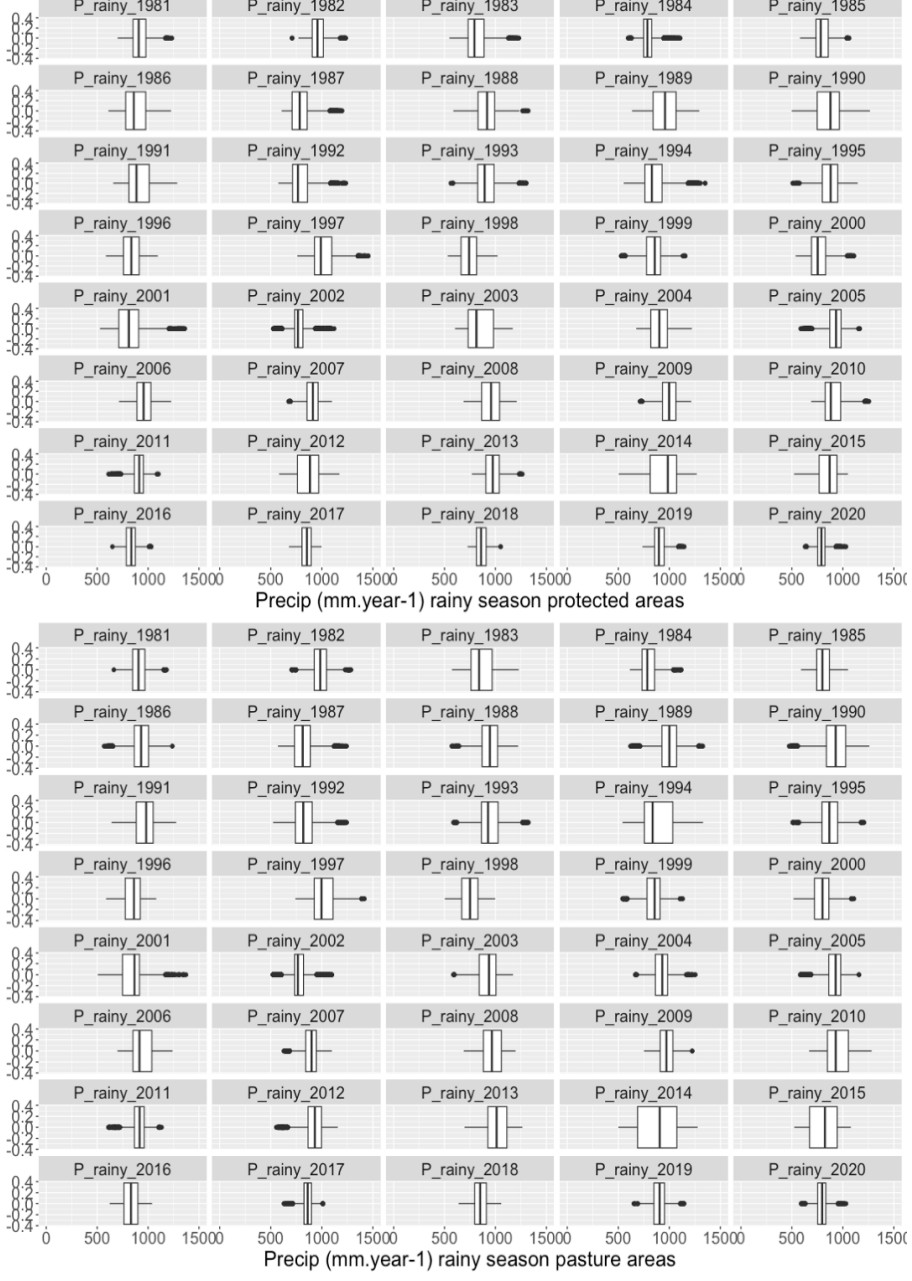

**Figure 8.** Yearly accumulated rainfall for the rainy season in Rondônia for the years 1981 to 2020 using CHIRPS data. These are the mean values of all valid pixels of CHIRPS images for protected and pasture areas.

Regarding the spatial distribution of precipitation, the southwestern and north regions of the State accumulate the larger area of protected primary forests, according to Figure 9, for the rainy season years 1983 to 1987, 1990 to 1996. 1998 to 2003, 2012 and 2015 to 2020. For the central region of the state, the most anthropized area, in the rainy season, the areas that present the lower values are 1984, 1985, 1987, 1992, 1994, 1998 to 2002 and since 2014, a decrease of precipitation in the north region of the State.

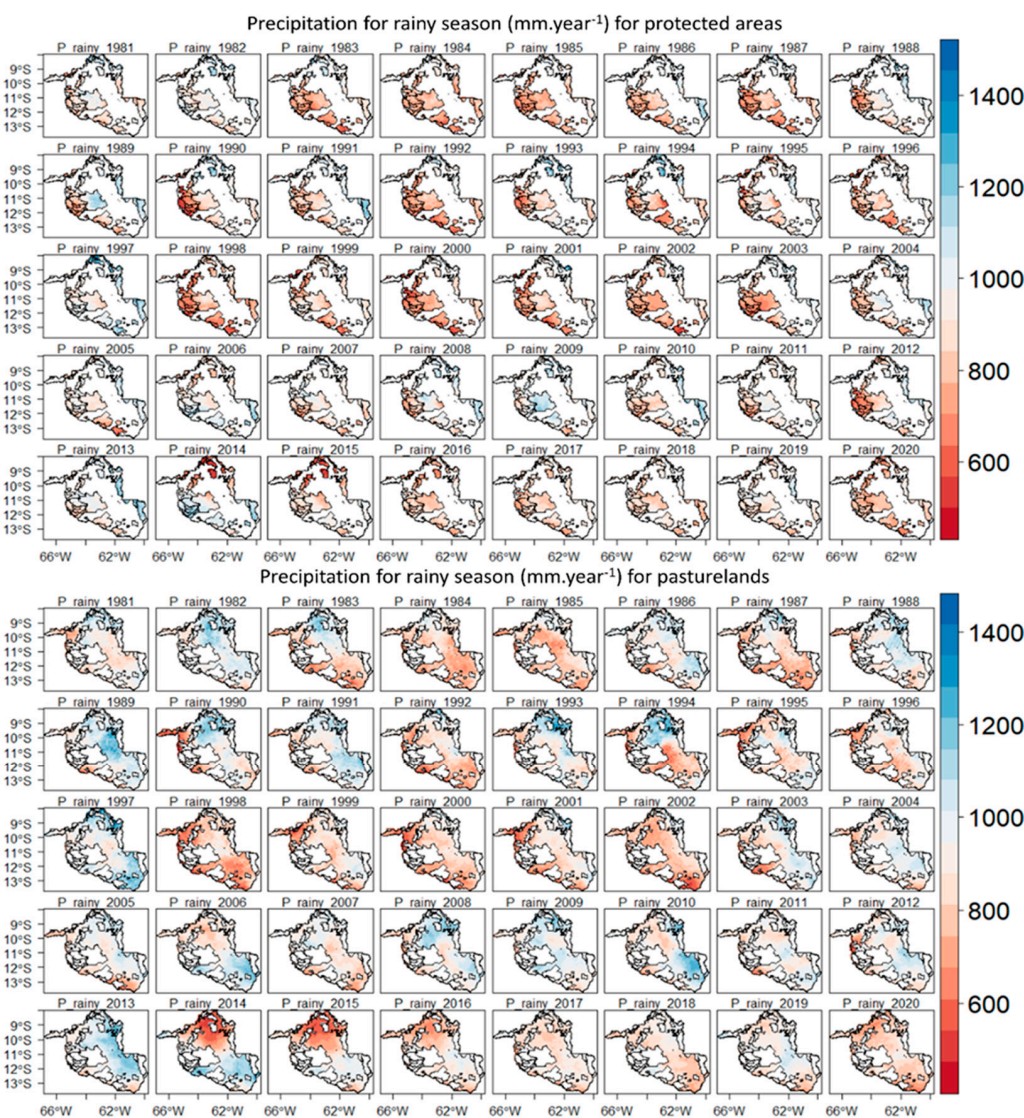

**Figure 9.** Spatialization of yearly accumulated rainfall for the rainy season in Rondônia for the years 1981 to 2020 using CHIRPS data for protected and pasture areas.

The Sen's slope measures the magnitude of change in a time series, presented in Figure 10. Here, it was calculated in a pixel-by-pixel scale and using CHIRPS annual accumulated data from 1981 to 2020. The results show no significant difference for the magnitude of change between Rondônia's protected areas and pasture areas. Pasture areas in the rainy season presented a higher reduction of precipitation than protected areas.

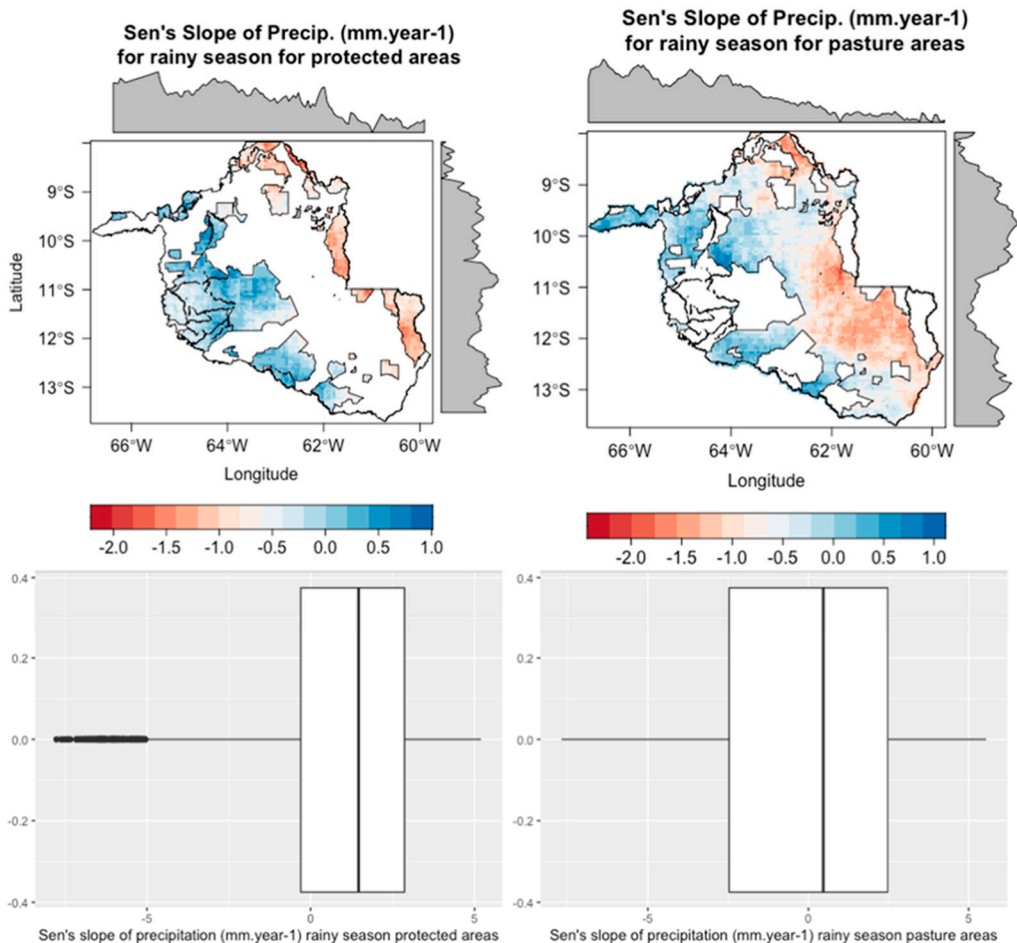

**Figure 10.** Spatialization of the Sen's Slope for rainfall for the dry season in Rondônia for the years 1981 to 2020 using CHIRPS data for protected and pasture areas.

3.4.2. Analysis of Precipitation in the Dry Season for Protected and Pasture Areas

In protected primary forests in the dry season, the years that presented lower values of rainfall with median values approximate to 100 mm.year$^{-1}$ were 1981, 1983, 1992, 1993, 1995, 2005, 2006, 2007, 2008, 2010, 2012 and 2015. For pasture areas in the dry season, the years that presented lower values of rainfall with median approximate to 100 mm.year$^{-1}$ were 1983, 1987, 1988, 1991, 1995, 1997, 1998, 2006, 2007, 2008, 2010, 2011, 2012 and 2015 (Figure 11).

Regarding the spatial distribution of precipitation, the southwestern and north regions of the State accumulate the larger area of protected primary forests, according to Figure 12, for the dry season, only years 1982, 1986, 2000 and 2016 did not present values below 250 mm.year$^{-1}$. For the north region of the state, in the dry season, the areas that present the higher values of precipitation were 1982, 1986, and 2018 (Figure 12).

The Sen's slope spatialization results show no significant difference for the magnitude of change between Rondônia's protected areas and pasture areas. Nevertheless, for the dry season, pasture areas presented a higher reduction of precipitation than protected areas (Figure 13).

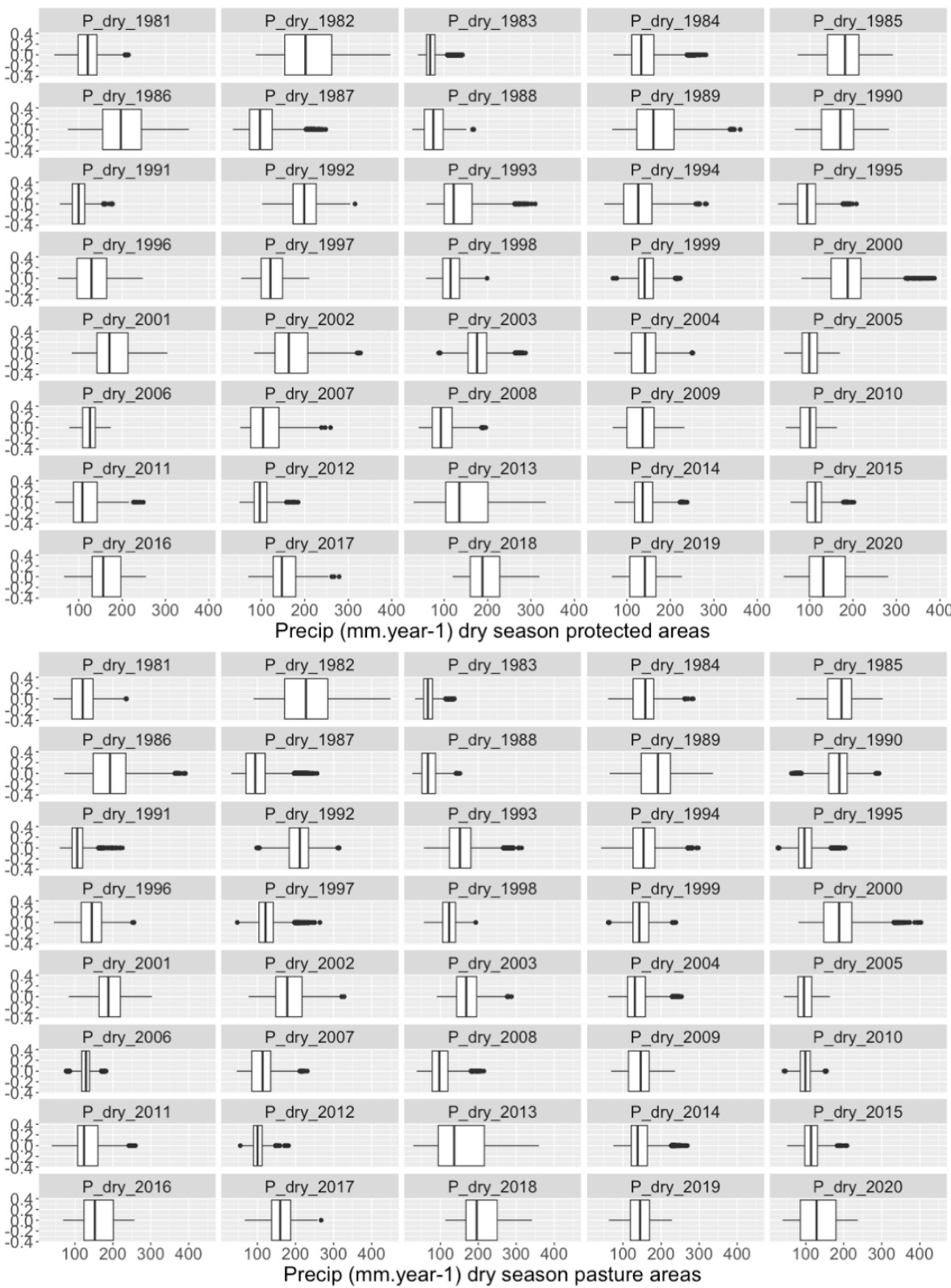

**Figure 11.** Boxplot of yearly accumulated rainfall for the dry season in Rondônia for the years 1981 to 2020 using CHIRPS data. These are the mean values of all valid pixels of CHIRPS images for protected and pasture areas.

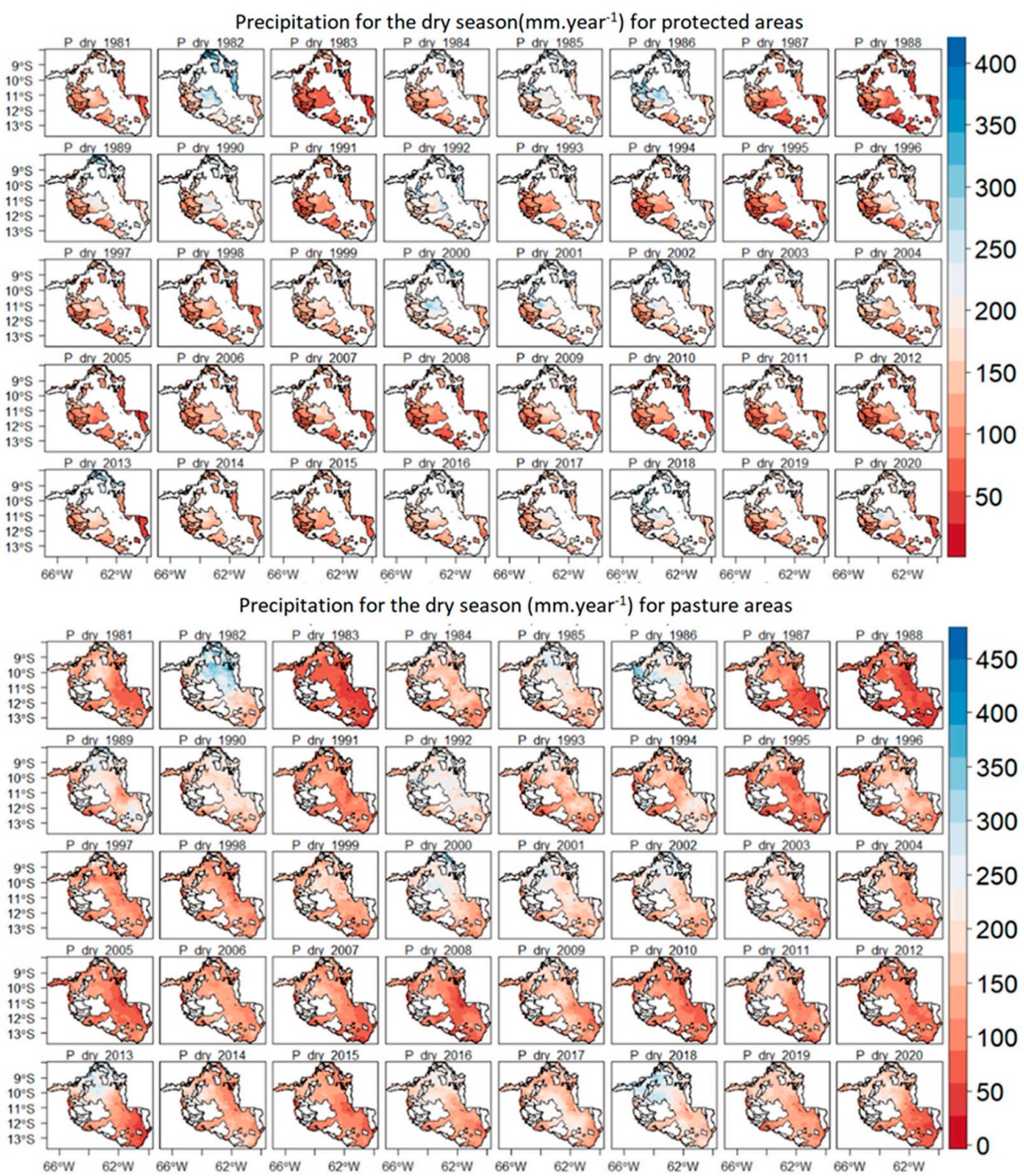

**Figure 12.** Spatialization of yearly accumulated rainfall for the dry season in Rondônia for the years 1981 to 2020 using CHIRPS data for protected and pasture areas.

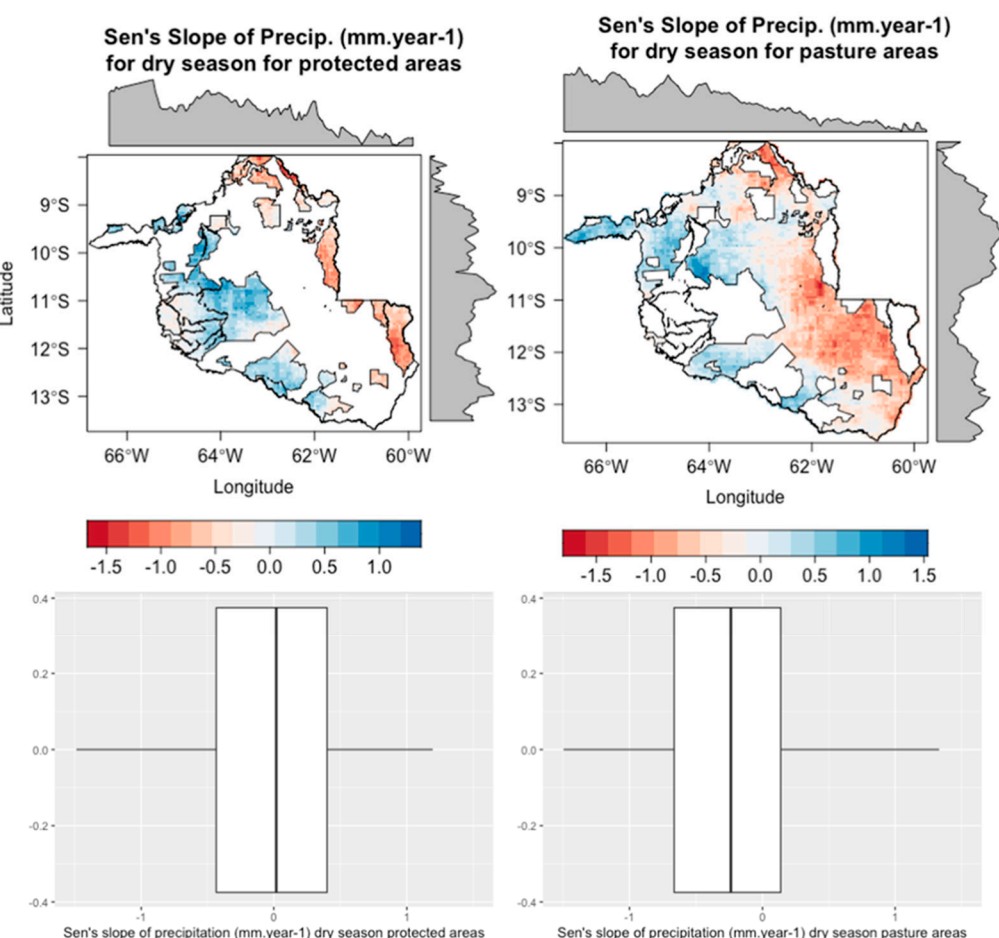

**Figure 13.** Spatialization of the Sen's Slope for rainfall for the dry season in Rondônia for the years 1981 to 2020 using CHIRPS data for protected and pasture areas.

Summary statistics of the raster layers are presented in Table 6.

**Table 6.** Descriptive statistics of precipitation (mm.year$^{-1}$) for years 1983, 1998, 2010, 2015 and 2020 presenting minimum, medium and maximum values for protected and pasture areas.

| Year | Dry Season | | | | | | | | Rainy Season | | | | | | | |
| | Protected Areas | | | | Pasture Areas | | | | Protected Areas | | | | Pasture Areas | | | |
| | min | med | max | CV | min | med | max | CV | min | med | max | CV | min | med | max | CV |
|---|---|---|---|---|---|---|---|---|---|---|---|---|---|---|---|---|
| **1983** | 39 | 67 | 142 | 0.25 | 33 | 66 | 136 | 0.26 | 547 | 794 | 1230 | 0.15 | 572 | 840 | 1229 | 0.16 |
| **1988** | 58 | 114 | 198 | 0.35 | 26 | 67 | 151 | 0.33 | 585 | 918 | 1321 | 0.13 | 571 | 945 | 1223 | 0.11 |
| **1998** | 26 | 74 | 168 | 0.23 | 56 | 122 | 193 | 0.18 | 530 | 740 | 1020 | 0.12 | 502 | 749 | 1173 | 0.13 |
| **2010** | 45 | 100 | 162 | 0.25 | 42 | 98 | 154 | 0.18 | 689 | 881 | 1255 | 0.13 | 669 | 824 | 1078 | 0.14 |
| **2012** | 50 | 96 | 184 | 0.21 | 54 | 100 | 181 | 0.15 | 580 | 883 | 1172 | 0.16 | 554 | 931 | 1155 | 0.11 |
| **2015** | 56 | 113 | 202 | 0.22 | 50 | 113 | 208 | 0.21 | 521 | 870 | 1050 | 0.16 | 523 | 824 | 1078 | 0.18 |
| **2020** | 40 | 132 | 281 | 0.35 | 39 | 128 | 237 | 0.37 | 630 | 790 | 1030 | 0.08 | 591 | 797 | 1030 | 0.08 |

Figure 14 summarizes the trend results of the Laplace test. Values above (+2) indicate increasing trends and values below (−2) indicate decreasing trends in precipitation over time. It is possible to observe the variability of the CHIRPS values (pixels) trends corresponding to the surface stations over the period from 1981 to 2020. It is noted between the series, that there were moments when the trends were within the values (+2 and −2) and

other moments positive trends with values above (+2) and for a long period trends with values below (−2) for the vast majority of series.

**Figure 14.** Laplace trend test spatialization for the State of Rondônia.

## 4. Discussion

Precipitation is key for regulating ecosystems and maintaining their equilibrium. Over the Amazon rainforest, El Niño-Southern Oscillation (ENSO) events have the potential to affect the forest's primary production [2]. During the El Niño year, tropical and subtropical latitudes experience higher temperatures. This impacts the precipitation regimens by decreasing the amount of rainfall for most of the tropical region of South America.

CHIRPS data tend to underestimate high values of monthly precipitation when compared to rain gauges [9]. This underestimation was also observed, for high monthly precipitation estimated by CHIRPS, by Paredes-Trejo et al. [49]. For the $R^2$ was lower in the western region of the State, also observed by Funk et al. [38] and Cavalcante et al. [9]. Other validation studies had similar results [1,2,49].

According to Peixoto [50] (p. 15), climate variability refers to time intervals to which a smaller defined cause can be associated and include the extreme values and the differences between annual, seasonal, monthly average values. According to the maps in Figure 7, there are some regions that presented years classified as wet (blue) or dry (red). The northern region of the State of Rondônia represents the lowest latitude, therefore, the hottest and wettest region, which probably contributes to the expressive volumes of precipitation. The periodicity of precipitation in the State is determined by the pattern of the atmospheric dynamics, such as the ITCZ, ZCAS, Bolivian High and ASAS, which act throughout the year. Therefore, these atmospheric dynamics characterizes a pattern in the rainfall rhythm with a wet season (October-April) and a dry season (May-September). In similar analyses, Zuffo; Franca [51] conclude that the spatial distribution of precipitation has lower rates in the southwest of the state and gradually increases in the east/southeast up to 2000 mm and even more in the north with values above 2500 mm.

Regarding the trends presented in Figure 8, to the southwest on the maps, a trend of increased precipitation is observed in protected areas, such as Federal Conservation Units and Indigenous Lands. However, it was noted a reduction in precipitation volumes in the flatter region with lower altitude to the north of the map. This decrease may be

associated with both topographic characteristics and changes in land use and occupation. The region demonstrates intense conversion from native forests to pasture, leading rainfall in recent years to be concentrated in more southwesterly areas of the State of Rondônia. This topographic configuration and land use influence the distribution and concentration of precipitation in the territory. These results are reinforced by Sen's slope (Figure 8), which is more negative in the north of the state, and positive (with an increasing trend) in protected areas with greater extension and restricted use.

Cavalcante et al. [9] analyzed 45 stations over the amazon and calculated 15 rainfall indices. The results presented by researchers conclude that 63 significant trends were detected using rain gauge data, of which only 13 were detected using CHIRPS products.

For the north region of the state, in the dry season, the areas that present the higher values of precipitation were 1982, 1986, and 2018. The values of precipitation for 1982, 1986, and 2018 years contrast the El Niño effect on the rainy season. These similar values were found by Li et al. [52] which states that negative rainfall anomalies are found in El Ninõ years in austral summer, when El Niño is mature. These results are corroborated by Panisset et al. [53] evidenced a precipitation deficit during the dry season for 2005 and 2010 drought events. Mu; Biggs and Shen [3] also provide evidence that these drought events presented lower values.

For the rainy and dry season, pasture areas presented lower values of precipitation than protected areas. Mu; Jones [24] developed an observational analysis of rainfall for the Brazilian Legal Amazon and related it to age of deforestation and the results show large spatial variability of trends. The dry season is driven and highly dependent by locally generated convection, increasing the effects of deforestation. Also, their results show a coherent relationship between negative dry-season precipitation trends and old-age deforested areas.

As indicated by the SS values, the zones with greater density of protected natural areas, such as Conservation Units and Indigenous Lands, in the southwest of the state, presented a trend of increasing precipitation. Areas with higher rates of conversion from native forest to pasture, in the north of the state, show a greater tendency to decrease. Given this scenario, it is important to point out that, with the exception of protected areas, part of the unprotected areas was deforested and replaced by pasture due to the expansion of the agricultural frontier towards the North Region of Brazil. The deforestation decreases the net surface radiation due to the increase in albedo, a consequence of the vegetation cover loss.

Deforestation can affect mechanisms and patterns of regional-scale precipitation, for the Rondônia case study, the shift from small-scale to large-scale deforestation have altered the regimen of precipitation by reducing rainfall in the dry season. The hydrological cycle and energy balance is directly affected by deforestation. They are affected by reduction on evapotranspiration, increasing in heat flux, decreasing the latent flux, increase in albedo reducing the absorbed solar radiation and decrease in surface roughness [54].

It is important to note that, with the exception of protected areas, part of these non-protected areas was converted to pasture throughout the rest of the state in 2021. The net surface radiation in tropical forests is mainly divided into latent heat flux, instead of sensible heat flux. This situation is explained by the dense vegetation and deep rooting systems. In similar analyses, Sales et al. [54] modeled scenarios to assess the influence of protected areas on climate variables. In the experiments, the sensible heat demonstrated the highest values in deforested areas and in the Southwest Region of the State. The latent heat flux presented the highest values in protected forest areas. In general, deforestation decreases the net surface radiation due to the increase in albedo as a consequence of native vegetation cover loss. And it directly influences the evapotranspiration processes, as well as the precipitation of water vapor that finds greater heat over pasture areas, concentrating, in certain scenarios, precipitation and not allowing it to reach areas of native vegetation in the Southeast.

The map in Figure 14 shows the general trend of monthly precipitation for the CHIRPS time series using the Laplace method. It is noted that the map show, in most of the location of surface stations, a negative trend of about 56% in monthly precipitation in the study area. On the other hand, about 25% of the data show an upward trend and about 19% within normal rainfall in the study area.

Negative rainfall trends with an average reduction in precipitation of 5 mm per year in the Legal Amazon were identified by [55], in the southern region of the Legal Amazon by [24] along the "deforestation arc" and in the southern region of the Amazon Basin by [10], where deforestation is widespread, evidenced a significant drought trend. For the authors, deforestation older than a decade increased rainfall in the dry season and older deforested regions reduced rainfall during the dry season. It is important to point out that a large number of points located in the northern region of the map showed a tendency towards a reduction in monthly rainfall for Rondônia. The points located in the central-eastern part showed a trend within the pattern and the points located in the western portion an increase in monthly rainfall. This pattern of negative trends was also observed in the region by [24] who pointed to decreasing trends in rainfall for the northern region and increasing trends for the western region of Rondônia. In view of this, monthly rainfall has been reducing by 56% for the state of Rondônia, mainly, a trend of reduction of 43% in the north region and 12% in the south of the state of Rondônia.

## 5. Conclusions

In general, the results obtained from the comparative statistical analysis between CHIRPS and observed precipitation datasets over Rondônia were satisfactory.

The analysis revealed that CHIRPS presents a tendency to underestimate precipitation values in most cases. Among the metrics, mean values between very good ($<\pm15\%$) and good ($\pm15$–$\pm35\%$) were observed using PBIAS; mean RMSE values range from 57.8 mm to 107.9 mm; an average agreement level of 0.9 and an average SES of 0.5; and good fit for the linear regression model (average $R^2 > 0.70$) for about 64.7% of the stations.

Among the analysis of distribution, classification and trends performed on historical rainfall data, a fluctuation in rainfall volumes was observed along the seasonality during the 40 years of records from the CHIRPS platform. For the seasonal analysis, the rainy season showed lower amplitudes in rainfall values and the transition seasons showed a constant pattern of reduction in total rainfall. The pattern presented in the pluviometric data demonstrates the seasonality that the tropical rainforest climate presents in the area. During the four decades of records, 2009 ranked as the wettest year (>3000 mm), and 2015 as the driest one (<1500 mm).

SPI–6 showed a predominance of the most recurrent dry condition in southwest areas, and a wetter one in areas to the north of Rondônia. The Northern Region of the territory has the lowest elevation and the lowest latitude, being, therefore, the hottest and wettest.

Finally, the analysis of trends through MK demonstrated negative trends for the months of January, November and July. For the months of May, August and October, it illustrated a positive trend. In addition, there was a spatial trend towards a reduction in the volume of rainfall in areas to the north of the State of Rondônia, and an increase in areas to the Southwest.

This difference may be associated with the effects generated by the process of transformation of land use and occupation over the decades, since in the Southwest Region there are areas of protected forests, such as Conservation Units and Indigenous Lands. The northern region, due to its constant change in land use and occupation, with intense conversion of areas from native forests to pasture, corresponds to the scenario observed in which rainfall, in recent years, is concentrated in areas further south-west of the State of Rondônia. The indications of these trends were reaffirmed with the metrics of Sen's slope.

In addition, it is possible to say that the information recorded by the rain gauges has a good relationship with the satellite estimates, which legitimizes the use of these data for rainfall analyses, especially for areas that do not have data available, such as the Amazon

region. The missing data that many rain gauges present when they do not continuously catalog precipitation or the difficulty of installing measuring instruments in the heart of the Amazon Forest is a serious problem.

The main challenges faced by CHIRPS is the superestimation and underestimation of precipitation for certain regions over the Amazon biome. This is mainly due the reduction of stations to calibrate values in more recent years. Future work are necessary to try to recalibrate the products using downscaling methods.

Therefore, these technologies and resources make it possible to fill in these gaps, providing a complete analysis of their series and, thus, contributing greatly to the development of research and to the understanding of the behavior of the atmosphere versus forest, especially in areas where human and financial needs make it difficult to collect data on a continuous and long-term basis.

**Author Contributions:** R.M.M. was responsible for conceptualization, writing, data acquisition, statistical analysis and preparing figures; B.C.d.S. and R.G.S. were responsible for writing, reviewing and preparing figures; F.d.S., V.B., S.S. and P.H.d.S. were responsible for writing and reviewing the manuscript. All authors have read and agreed to the published version of the manuscript.

**Funding:** This research received no external funding.

**Data Availability Statement:** The results/data/figures in this manuscript have not been published elsewhere, nor are they under consideration (from you or one of your Contributing Authors) by another publisher. I have read the MDPI journal policies on author responsibilities and submitted this manuscript in accordance with those policies. All of the material is owned by the authors and/or no permissions are required. Data will be made available upon request.

**Acknowledgments:** The publication of this article was funded by the Open Access Fund of the Leibniz Association.

**Conflicts of Interest:** The authors declare no conflict of interest.

**Ethical Approval:** All authors have read, understood, and have complied as applicable with the statement on "Ethical responsibilities of Authors" as found in the Instructions for Authors and are aware that with minor exceptions, no changes can be made to authorship once the paper is submitted. I confirm that I understand the MDPI's Climate journal is a transformative journal. When research is accepted for publication, there is a choice to publish using either immediate gold open access or the traditional publishing route.

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
