# Peer review of "Precipitation Variability for Protected Areas of Primary Forest and Pastureland in Southwestern Amazônia"

_climate, doi:10.3390/cli11020027_

Round 1

Reviewer 1 Report

Analysis of spatiotemporal variation of precipitation using CHIRPS and rain gauges for primary forest and deforested areas in southwestern Amazônia

Thank you for providing me chance to review this manuscript. I am glad to review this manuscript because it’s very interesting research but there are many flaws in this research. Figures are well appropriate according to journal criteria. The author did spatiotemporal analysis of the precipitation distribution for Rondônia State, in southwestern Amazonia. They analyzed data from Climate Hazards Group InfraRed Precipitation and Station (CHIRPS), using a pooled time analysis of a forty-year period (1981-2020). Data obtained from remote sensing were validated by rain gauges distributed over the study region.

This research needs a lot attention to revise this manuscript.

Major and minor comments are attached below

Abstract section is not appropriate. The author should revise the whole abstract section

There is no continuity in the paragraphs. Secondly the author added many small paragraphs in the introduction section.

Objectives are not clear at the end of the introduction section.

Materials and Methods

The author should add the heading of study area or location of study area and then explain the whole study area in the study area section.

Selection of rain gauges and collection of CHIRPS data

The author should split both sections.

Standard Precipitation Index (SPI-6)

The author should add equation number in in the whole manuscript.

Results

Agreed

Discussion

Agreed

Conclusion

Agreed

I hope the author will modify as soon as possible then they will resubmit revised manuscript in very good form.

Best of luck

Author Response

Reviewer 1

Reviewer

Comments from reviewer

Response to reviewers

Reviewer 1

Abstract section is not appropriate. The author should revise the whole abstract section

Abstract was adequated, the text presenting findings was added: "Validation metrics presented good agreement between CHIRPS estimations and rain gauges observations. The pasture areas for the dry season presented larger areas for decreasing values for the Sen’s Slope. According to the Standard Precipitation Index, years 1983, 1998, 2010 and 2015 are the driest years, reflecting effects of El Niño, with lower values and longer dry periods for pasture areas."

There is no continuity in the paragraphs. Secondly the author added many small paragraphs in the introduction section.

The introduction section was re-written, we focused on fluidity of the text and robustness of discussions, avoiding short paragraphs.

Objectives are not clear at the end of the introduction section.

The following text was added at the end of the Introduction section: "The objective of the researchers was to present a spatial-temporal trend analysis of precipitation for primary forest protected areas and non-protected areas."

The author should add the heading of study area or location of study area and then explain the whole study area in the study area section.

We have adequated and added the following paragraphs: 'During the summer, atmospheric circulation exhibits an important pattern through the trade winds that transport moisture from the Tropical Atlantic to the Amazon region. Added to this, at this time of the year, due to the advances of cold fronts coming from the southern portion of Brazil, a thermal low over the Chaco region, are associated with intense convective activities, increasing cloud cover over the Central Amazon region and , therefore, providing conditions for the SACZ to be more active and intense on rainfall in the southern and western regions of the Amazon.
Rainfall seasonality is an important feature, as it allows understanding of interannual variability and extreme rainfall events. Changes in these rainfall patterns can cause impacts due to natural causes or anthropic processes. In this sense, interannual variability depends on the fields of sea surface temperature (SST) anomalies in the Pacific or Tropical Atlantic at the beginning and end of the rainy season. The combination of anomalous atmospheric circulations, influenced by SST, affect the positioning of the ITCZ over the Atlantic and, therefore, influencing the rainfall distribution over northern South America. Therefore, El Niño and La Niña events cause an increase or decrease in rainfall anomalies in the Amazon.
According to Butt et al. (2011), the increase in deforestation in recent decades in the southern region of the Amazon has generated some changes in land use and occupation, which may influence the transition period from the dry to the wet season in Rondônia, due to the decrease in forest cover in the region. region, where changes in surface radiation balance caused by deforestation are the likely drivers of changes in rainfall patterns in the rainy season. In this sense, although changes in annual precipitation totals are not significant, the implications of a longer dry season are important for a variety of benefits that forest cover exerts, such as water availability and ecological aspects.'

The author should split both sections. Selection of rain gauges and collection of CHIRPS data

We have splitted as requested.

The author should add equation number in in the whole manuscript.

Reviewer 2 Report

Review for "Analysis of spatiotemporal variation of precipitation using CHIRPS and rain gauges for primary forest and deforested areas in southwestern Amazônia". The manuscript examined the validation of CHIRPS in southwestern Amazônia based on 17 rain gauge. The manuscript is generally moderate and the topic seems to present few interesting results for readers. Overall, I suggest a "major" revision before possible consideration of the application in Climate. It has some research implications.However, the following issues require attention:

1. The title is "Analysis of spatiotemporal variation of precipitation using CHIRPS and rain gauges for primary forest and deforested areas in southwestern Amazônia". I think the title is misleading and it should be revised carefully. 

2. Although the manuscript is generally well-written, a language check by a professional native speaker or an editing agency is needed to fix some syntax, style, and phrasing problems.

3. The Introduction section must be enhanced a lot! The introduction needs to be improved and further discussion is needed. Also, please use either rainfall or precipitation in the manuscript. Also, introduction required adding newly references.

4. Use KGE metric as it can evaluate multiple properties together in one integrated metric. It combines Pearson's correlation (r), the ratio of spatial variability and the normalized difference.

5. Please revise the Abstract to be a mirror of the whole paper. It requires some sentences about your findings.

6. The Discussion section needs to be improved using the above-mentioned papers.

7. Please improve the conclusion with your findings.

8. I want to understand why the authors only used CHIRPS dataset?

9. Please add more literature review about: similar studies in same region of nearby areas. 

10. Instead of using Slope linear trend test, authors is suggested to use Sens’s slope and modified Mann-Kendall to check the significant.

11. In figure 1, please use the same coordinate systems. 

12. In Table 1, no need for identification code and please add spatial distribution map for the stations.

13. Please center Figure 3 in the middle of the page.

14. Figure 5 is too small. Please increase the size of the figure.

15. What is the station number used for Table 5

16. Figure 6: please show only statical significant (P<0.05) for the significant test.

17. Figure 9: merge them in one figure.

18. Authors should not write small paragraphs (2 to 3 lines).

19. Please check declaration section 

20. Please check the references as there are many references without Doi. Please check them carefully.

I look forward to seeing a better version of the manuscript.

Author Response

Reviewer 2

Reviewer

Comments from reviewer

Response to reviewers

Reviewer 2

The title is "Analysis of spatiotemporal variation of precipitation using CHIRPS and rain gauges for primary forest and deforested areas in southwestern Amazônia". I think the title is misleading and it should be revised carefully.

We adequated the tittle for: 'Precipitation variability for protected areas of primary forest and pastureland in southwestern Amazônia'

Although the manuscript is generally well-written, a language check by a professional native speaker or an editing agency is needed to fix some syntax, style, and phrasing problems.

One of the co-authors is a native speaker and have checked the manuscript.

The Introduction section must be enhanced a lot! The introduction needs to be improved and further discussion is needed. Also, please use either rainfall or precipitation in the manuscript. Also, introduction required adding newly references.

Section adequated.

Use KGE metric as it can evaluate multiple properties together in one integrated metric. It combines Pearson's correlation (r), the ratio of spatial variability and the normalized difference.

The KGE metric was implemented in the manuscript, we thank the reviewer for the suggestion.

Please revise the Abstract to be a mirror of the whole paper. It requires some sentences about your findings.

Abstract revised.

The Discussion section needs to be improved using the above-mentioned papers.

Section adequated. The papers that the reviewer mentioned did not reached the authors. We could not find them in the review.

Please improve the conclusion with your findings.

I want to understand why the authors only used CHIRPS dataset?

CHIRPS is the state-of-the-art dataset for precipitation estimates with daily availability for the last 40 years.

Please add more literature review about: similar studies in same region of nearby areas.

References added.

Instead of using Slope linear trend test, authors is suggested to use Sens’s slope and modified Mann-Kendall to check the significant.

The discussion regarding this requirement from the reviewer was adequated.

In figure 1, please use the same coordinate systems.

Figure was adequated.

In Table 1, no need for identification code and please add spatial distribution map for the stations.

The authors did not understood what the reviewer meant by "add spatial distribution" on a table.

Please center Figure 3 in the middle of the page.

Figure was adequated.

Figure 5 is too small. Please increase the size of the figure.

Size increased

What is the station number used for Table 5

We used the mean value for all pixels for the State of Rondônia.

Figure 6: please show only statical significant (P<0.05) for the significant test.

The authors discussed and decided to show all the values distributed in a continuous form. We thank the reviewer for the suggestion, though.

Figure 9: merge them in one figure.

Figures were merged according to the request.

Authors should not write small paragraphs (2 to 3 lines).

Small paragraphs were rewritten.

Please check declaration section

Section was adequated

Please check the references as there are many references without Doi. Please check them carefully.

DOI was added.

Reviewer 3 Report

Comments to the authors (if any)

I am writing regarding Manuscript climate-2098248B entitled "Analysis of spatiotemporal variation of precipitation using CHIRPS and rain gauges for primary forest and deforested areas in southwestern Amazônia" which was submitted to the Climate MDPI.

The current study reports an interesting topic that points out a spatiotemporal variation of precipitation using CHIRPS and rain gauges in an interesting area. The manuscript shows good scientific quality and originality in the studied region. Generally, I found the paper not potentially suitable for publication at its present stage. To reconsider my decision, the paper needs a major revision to make it well for publishing.

Comments

The abstract

1-      Please improve this sentence

Pixel-by-pixel trend analyzes were developed by applying the Mann-Kendall test. The spatialization of the precipitation’s tendency to increase or decrease was performed by using Sen’s slope method at a pixel-by-pixel scale

2-      Please improve this sentence by adding the main finding of the study and the results of the validation model errors.

Validation metrics showed good agreement between CHIRPS data and data provided by rain gauges. Sen’ slope specialization results show a  reduction of approximately -15 mm year-1, with a decrease mainly in the Northern Region of  Rondônia, which has extensive areas where the native forest has been replaced by pasture.

The introduction

1-      Lines 40-45, this part needs to be improved, please discuss in detail five papers at least.

2-      Lines 46-53: this part needs to be improved, please discuss in detail four papers at least (current and past climate variability of the Amazon Basin.

3-      Lines 65-68: please remove them.

4-      Lines 57-79: need to be improved by adding and discussing the application of CHIRPS data.

5-      Lines 80-86: please add the research gap.

Materials and Methods

1-      Please a subtitle (study area)

2-      The study area and the figures have been well presented

3-      Table 1, please add the statistical rainfall characteristics as the mean, max and variation coefficient

4-      The metrics for validating CHIRPS data have been well presented

5-      Add the number of all equations and cite them on all the paper

6-      Sen's slope analysis, add the presentation of each parameter of the equation (SS)

Results

1-      Two maps are needed, the first one includes the real in-site observation and the second one explains the spatial distribution of CHIRPS data of the study with an interpolation mapping technique.

2-      I suggest IDW for rainfall mapping

3-      Figure 3 is not clear, why the authors do not apply a spatial mapping technique as IDW to the various obtained results (R2, PBIAS (%), NSE, RMSE (mm), Willmott coefficient (d)).

4-      Please use IDW for updating the maps of figure 3.

5-      More than 45 % of validation results provide unsatisfactory NSE values, which makes the use of the proposed CHIRPS data as a management tool for the study area very not appropriate. How the authors continued the application of CHIRPS data and how to justify this unsatisfactory. This is the critical point of this paper.

6-      There is a need to add the characteristics of statistical parameters of observed and simulated rainfall such as mean, maximum (Max), minimum (Min), standard deviation (STD), and coefficient of variation (CV)

7-      For Tables 4 and 6. Some previous comment.

8-      Table 5. We need all results of MK and Sen’s tests.

9-      3.4.1 Spatial analysis of precipitation trends in the State of Rondônia separated per dry and rainy season for anthropized and primary forest areas. This section was well discussed and referenced.

Conclusion

1-      The unsatisfactory validation model using CHIRPS data must be focused on in this section by suggesting other solutions and recommendations

2-      What are the strengths, and weaknesses of this research?

There are several comments but I will stop here and see what the authors do in the revised version to continue the review process.

Reviewer 4 Report

The articles entitled “Analysis of spatiotemporal variation of precipitation using CHIRPS and rain gauges for primary forest and deforested areas in southwestern Amazônia” presents an important contribution to the study area using surface data with those estimated and modeled by orbital platforms.

General comments:

In addition to the platforms indicated in the text (CHIRPS, TRMM, GCMM and CMORPH) there are other platforms such as CHELSA (https://chelsa-climate.org/) that could be indicated as data sources.

We recommend including the reference (Alvares CA, Stape JL, Sentelhas PC, et al (2013) Köppen's climate classification map for Brazil. 478 Meteorologische Zeitschrift 711–728. https://doi.org/10.1127/0941-2948/2013/ 0507) in characterizing the study area, as well as evaluating the data available at: http://www.cprm.gov.br/publique///Mapas-e-Publicacoes/Atlas-Pluviometrico-do-Brasil-1351.html

In the comparative analysis between the observed and estimated data, you could add the Laplace trend test and also the Mean Error (ME) test.

Figure 3 needs to be expanded as it is not possible to visualize the data spatialization in detail.

What is the explanation for this higher or lower ME value in the study area? Could it be associated with a higher occurrence of rain gauges in certain areas of the state?

What is the coverage area of ​​rain gauges in the context of the study area?

Why adopt the administrative limit (State of Rondonia) as a study area? Why not a watershed or other spatial boundary?

We believe that the first map to be presented is the one that expresses the spatial and temporal/monthly distribution of rainfall in the study area.

In Figure 5, I believe it is more appropriate to represent the correlation coefficient (R) that varies from 1 to -1. I think this would better express the spatial correlation of the data, even indicating the places with positive and negative correlation.

In some moments in the text the series is indicated as having 40 years of duration in another 38 years. To review. This figure needs a better resolution (300 or 600 dPis).

In general, the illustrations need enlargement. From the way they are presented, it is difficult to visualize what is described in the text.

A characterization of the major climate systems that current in the study area should be incorporated into the text.

What is the purpose of evaluating data in pasture areas and protected/preserved areas? Do the authors believe that the results will be different depending on the different types of land use?

It is important to define which types of conservation/protection unit the authors refer to in the text: integral protection or sustainable use.

Author Response

Reviewer 3

Reviewer

Comments from reviewer

Response to reviewers

Reviwer 3

In addition to the platforms indicated in the text (CHIRPS, TRMM, GCMM and CMORPH) there are other platforms such as CHELSA (https://chelsa-climate.org/) that could be indicated as data sources.

Added

We recommend including the reference (Alvares CA, Stape JL, Sentelhas PC, et al (2013) Köppen's climate classification map for Brazil. 478 Meteorologische Zeitschrift 711–728. https://doi.org/10.1127/0941-2948/2013/ 0507) in characterizing the study area, as well as evaluating the data available at: http://www.cprm.gov.br/publique///Mapas-e-Publicacoes/Atlas-Pluviometrico-do-Brasil-1351.html

Recommendation accepted and adequated the map using the Koppens climate classification and added the climatologic normal from the National Institute of Meteorology.

In the comparative analysis between the observed and estimated data, you could add the Laplace trend test and also the Mean Error (ME) test.

The recommended analysis were implemented in the paper, we thank the reviewer for the suggestion.

Figure 3 needs to be expanded as it is not possible to visualize the data spatialization in detail.

Figure was adequated.

What is the explanation for this higher or lower ME value in the study area? Could it be associated with a higher occurrence of rain gauges in certain areas of the state?

The explanation was added to the text and it reads: ""

What is the coverage area of ​​rain gauges in the context of the study area?

We have 16 stations in the State that meet the criteria, as can be seen in text: "Rainfall gauge data was obtained using the National Water Agency (ANA) plugin for the QGis program [34]. The ANA Data Acquisition tool automatically downloads from several pluviometric and fluviometric stations. Using the boundary of the state of Rondônia as an area of interest, 162 stations were first selected. After a first filtering, considering a temporal criterion, only 28 rain gauges had data from 1981 to 2020. This interval was defined due to the CHIRPS data being available since 1981. A new filtering was performed on these 28 stations, selecting only those presenting less than 15% of missing data. After these filtering steps, the researchers selected 16 stations, shown in Table 1."

Why adopt the administrative limit (State of Rondonia) as a study area? Why not a watershed or other spatial boundary?

Discussion explaining the reason for selection of the study area is described, the addedd text reads: "We selected the geopolitical boundary due to the State's lack of studies regarding trends in precipitation over protected and non-protected areas. The State of Rondônia is a great case study for deforestation, since the 1980's, when we have the start of Landsat 4 mission, it is notable the conversion of primary forests to pasture."

We believe that the first map to be presented is the one that expresses the spatial and temporal/monthly distribution of rainfall in the study area.

A map with this recommendation was developed and can be seen in figure 1.

In Figure 5, I believe it is more appropriate to represent the correlation coefficient (R) that varies from 1 to -1. I think this would better express the spatial correlation of the data, even indicating the places with positive and negative correlation.

Figure was adequated.

In some moments in the text the series is indicated as having 40 years of duration in another 38 years. To review. This figure needs a better resolution (300 or 600 dPis).

The years studied was corrected to 40 years.

In general, the illustrations need enlargement. From the way they are presented, it is difficult to visualize what is described in the text.

Illustrations were adequated.

A characterization of the major climate systems that current in the study area should be incorporated into the text.

What is the purpose of evaluating data in pasture areas and protected/preserved areas? Do the authors believe that the results will be different depending on the different types of land use?

Our hypothesis was that precipitation in these areas would be different, but as results show, they are different but with no statistical significance. Also, variables such as albedo, roughness and latent and sensible heat differ from both types of land covers.

It is important to define which types of conservation/protection unit the authors refer to in the text: integral protection or sustainable use.

The protected areas are a mix of integral protection and sustainable uses, since we used the vectorial boundaries available on ICMBio website. We did not differentiated.

Round 2

Reviewer 1 Report

I found still there are minor mistakes in equations numbering and formatting. 

The author should prepare figures and tables according to the journal format.

There are many figures. The author add some figures in the supplementary data.

Best of luck

Reviewer 2 Report

The new version of the manuscript can be accepted now 

Reviewer 3 Report

After the review process of The paper (climate-2098248)    The authors have significantly  impove the paper after the first round of revision.  The paper is accepted in the current form.